# Almost Bayesian: Dynamics of SGD Through Singular Learning Theory

**Max Hennick**
Department of Mathematics and Statistics
University of New Brunswick & TrojAI
mhennick@unb.ca

**Stijn De Baerdemacker**
Department of Mathematics and Statistics, Department of Chemistry
University of New Brunswick stijn.debaerdemacker@unb.ca

## Abstract

The nature of the relationship between Bayesian sampling and stochastic gradient descent in neural networks has been a long-standing open question in the theory of deep learning. We shed light on this question by modeling the long runtime behaviour of SGD as diffusion on porous media. Using singular learning theory, we show that the late stage dynamics are strongly impacted by the degeneracies of the loss surface. From this we are able to show that under reasonable choices of hyperparameters for SGD, the local steady state distribution of SGD is effectively a tempered version of the Bayesian posterior over the weights which accounts for local accessibility constraints. We then empirically verify the porous diffusion picture across multiple models and datasets, and provide experimental evidence of the steady state-Bayesian posterior correspondence.

## 1 Introduction

One of the core problems in developing a scientific theory of deep learning models is giving a descriptive theory of how the internal model structure evolves during training as the model gains "knowledge" about its training distribution ((McGrath et al., 2022), (Olsson et al., 2022)) and how this evolution relates to the generalization ability of deep learning models. Classical methods for understanding model generalization such as the Bayesian Information Criterion (Schwarz, 1978) fail to give accurate descriptions of the generalization behavior of deep learning, due to its "singular" nature (Wei et al., 2023).

This has lead recent research to utilize Watanabe's *singular learning theory* (SLT) (Watanabe, 2009) as the basis for studying deep learning models. The key result of singular learning theory is the *widely applicable Bayesian information criterion* (Watanabe, 2012) which (broadly speaking) says that the generalization error of a model with parameter $w$ is controlled by the *learning coefficient $\lambda(w)$*, which corresponds to the "complexity" of some local area around the parameter. Measuring how this quantity evolves over time has been proposed as a method to study the emergence of structure within neural networks ((Lau et al., 2024), (Wang et al., 2024a)) and has given very promising results.

Despite this, it is not clear how the dynamical picture of SGD interacts with the purely Bayesian description of SLT. It has been shown that there is seemingly some relationship between Bayesian sampling of parameter space of neural networks and SGD, both experimentally (Mingard et al., 2020), and theoretically under assumptions of non-degeneracy of minima of the loss (Mandt et al., 2016b) (which is known to be false in general). Here we extend this connection to the more general case by describing the late stage training dynamics of SGD using a fractional Fokker-Planck equation which can be solved explicitly under reasonable assumptions. We show that the steady-state solution of this equation is related to the purely Bayesian posterior by tempering probabilities based on accessibility constraints determined by the learning coefficient. Potential practical applications of the results presented here are discussed in appendix C.

## 2 RELATED WORK

### 2.1 SINGULAR LEARNING THEORY

Our work relies upon results coming from singular learning theory ((Watanabe, 2012), (Watanabe, 2022), (Watanabe, 2024), (Watanabe, 2009)), the known relationship between inference and thermodynamics (LaMont & Wiggins, 2019), and the application of singular learning theory to the study of deep learning, referred to as *developmental interpretability* ((Wang et al., 2024b), (Wang et al., 2024a), (Chen et al., 2023)). We make particular use of the estimation methods for the local learning coefficient introduced in (Lau et al., 2024) using (van Wingerden et al., 2024).

### 2.2 GRADIENT NOISE AND SGD DYNAMICS

The methods used here are related to the *Stochastic Gradient Noise model* (SGN) of SGD ((Zhou et al., 2021), (Battash & Lindenbaum, 2023), (Nguyen et al., 2019), (Simsekli et al., 2019), (Mignacco & Urbani, 2022)) due to the relationship between the Fokker-Planck equation and the Langevin equation used in SGN. This framework has been used, for example, to model escape times from local minima (Xie et al., 2021).

Other works have studied the diffusive-like dynamics of SGD (Fjellström & Nyström, 2022), and even modeled SGD as an Ornstein-Uhlenbeck process to relate the dynamics of SGD back to the purely Bayesian case (Mandt et al., 2016b). However, this framework requires that the minima of the loss be quadratic, which means it cannot accurately capture the behaviour of SGD in neural networks due to the degeneracy of local minima. Furthermore, other works have also found connections between SGD and fractal geometry ((Camuto et al., 2021), (Şimşekli et al., 2021)) by the use of iterated function systems and Feller processes. Although related to the results here, the formalisms used are significantly different and the exact relationship is not straightforward.

The most closely related to the work here is (Chen et al., 2021) who show that many networks seem super-diffusive near initialization and decay into sub-diffusion over time. They also give a relationship to a type of fractal diffusion to explain this. However, they give no theoretical results, relying entirely on experimental results to draw conclusions. Our work instead focuses on a rigorous theoretical model that allows us to develop a theory about the long runtime nature of SGD which explains the observations made previously, and we provide experimental results to verify theoretical predictions.

## 3 FRACTIONAL DYNAMICS OF DEEP LEARNING

### 3.1 GRADIENT NOISE AND THE FOKKER-PLANCK EQUATION

Consider a neural network defined by some set of parameters $w \in W$ (where we assume $W$ is compact throughout) and let $\mathcal{X}$ be the set of tuples $(x_i, f(x_i))$ where $f$ is the oracle that describes our decision problem. Denote the loss function by $L : \mathcal{X} \times W \to \mathbb{R}$ and set $\mathcal{L}[\mathcal{X}, w] = \mathbb{E}_{\mathcal{X}}[L(x, w)]$. Letting $X_m \subset \mathcal{X}$ be a randomly sampled subset of possible inputs, the empirical loss on $X_m$ will then be denoted $\mathcal{L}_m[X_m, w] = \mathbb{E}_{X_m}[L(x, w)]$. For the purposes of the theoretical analysis, we will assume that we are working in the large batch size regime so that the estimation noise of the loss (and gradient) doesn't dominate the dynamics of the system.

#### 3.1.1 GRADIENT NOISE AND SUB-DIFFUSION

There is extensive literature which attempts to capture the dynamics of SGD by decomposing the weight updates (under some abuse of notation) into the form

$$\frac{dw}{dt} = -\gamma \nabla \mathcal{L}(w_{t-1}) + \Sigma_{w_{t-1}} \tag{1}$$

where $\mathcal{L}$ is the population loss, $t$ is the timestep, $\gamma$ is the learning rate, and $\Sigma_{w_{t-1}}$ is a random vector (which we will assume in this work is an anisotropic Gaussian). This is what is generally referred to as a *Langevin stochastic differential equation*. Systems governed by such SDEs have a displacement $R(t) \propto t^{\frac{1}{2}}$, meaning they diffuse like Brownian motion.

However, most works which examine the weight dynamics don't agree with this model. It has been found that networks trained under SGD can behave super-diffusively early in training, becoming sub-diffusive as training continues (Chen et al., 2021). Our experiments agree with this, finding that the displacement of neural network weights after long run times are described well by a power law like $R(t) \propto t^{\frac{1}{\nu}}$ for $\nu \geq 2$ (Bouchaud & Georges, 1990) (an example of which can be seen in figure 1). Similarly, it has been observed that the weight movement of SGD (with momentum and weight decay) can have weight displacement that scales logarithmically like $R(t) \propto \ln t$ (Hoffer et al., 2018). This behaviour cannot be captured by the traditional Langevin equation and requires the introduction of a *subordination* term. Such non-Brownian diffusion is collectively referred to as *anomalous diffusion*.

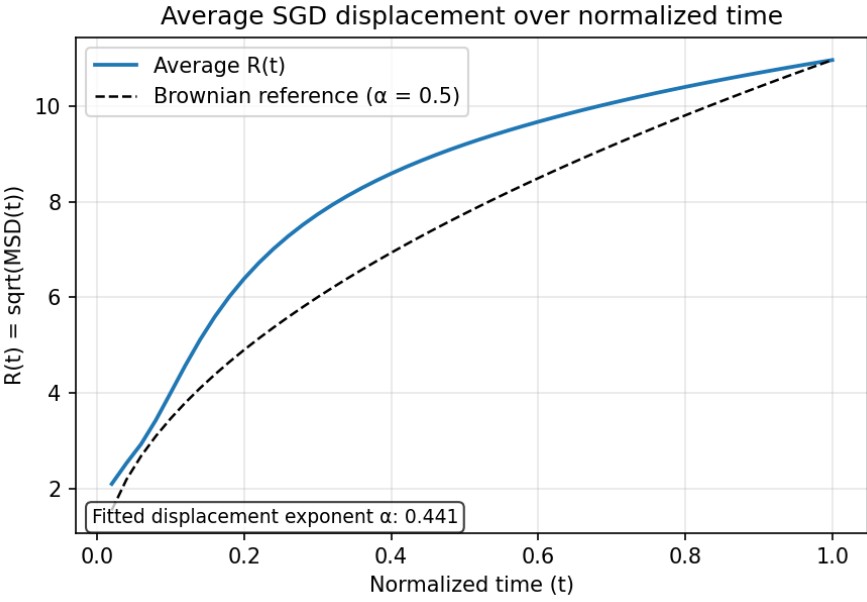

Figure 1: Mean weight displacement of a collection of fully connected neural networks trained using SGD on a randomly generated Moons dataset (Pedregosa et al., 2011), compared with expected displacement in the case of Brownian motion. It can be seen that this displays anomalous diffusion corresponding to early super-diffusion followed by late stage sub-diffusion.

To tackle this problem, we move into a formalism which is dual to the SDE picture, being the Fokker-Planck equation. Intuitively, the SDE picture describes the stochastic evolution of a single run while the Fokker-Planck equation is what describes the deterministic evolution of the probability distribution over parameter space over time determined by the SDE.

We now give the Fokker-Planck equation (FPE) in weight space (that is, $\nabla = \nabla_w$):

$$\frac{\partial p(w,t)}{\partial t} = \nabla \cdot (D(w,t)\nabla p(w,t) - \gamma p(w,t)\nabla \mathcal{L}(w)) \tag{2}$$

where $p$ is a probability density function (density of states), $D$ is the diffusion coefficient, $\gamma$ is a scalar (usually called friction), and $\mathcal{L}$ is a loss function which in a physical sense acts as a potential energy.

While the standard FP equation describes the behaviour of standard Brownian motion, one can hande the sub-diffusive case by introducing the *(Caputo) fractional derivative operator*(Diethelm, 2019) $\mathcal{D}_t^\alpha$ where $0 < \alpha < 1$ is a real number. Letting $f$ be some arbitrary (differentiable) function of $t$ the Caputo fractional derivative operator is defined as

$$\mathcal{D}_t^\alpha f(t) = \frac{1}{\Gamma(1-\alpha)} \int_0^t \frac{f'(t)}{(t-\tau)^\alpha} d\tau \tag{3}$$

We now define the (time) fractional Fokker-Planck equation (FFPE) for SGD[1] as:

$$\mathcal{D}_t^\alpha \, p(w,t) = \nabla \cdot (D(w,t)\nabla p(w,t) - \gamma p(w,t)\nabla \mathcal{L}_m[w]) \qquad (4)$$

Where the $X_m$ is dropped from the loss expression for simplicity. We note here that this equation itself does not directly describe ultra-slow diffusion (that is, where the displacement $R(t) \propto \ln t$). However, ultra-slow diffusion appears "in the limit" of power law sub-diffusion (Kochubei, 2008). A discussion of this as well as the role of the fractional derivative are given in in the appendix.

One might now try to modify this to a "time-space fractional" Fokker-Planck equation to account for the potential super-diffusive behaviour early in training. However, we are interested in studying the steady states of the system, and under some very mild assumptions (namely that the probability distribution doesn't lose mass over training) the steady state of the system does not depend on this early stage of training, and can be captured by solving the given time fractional FPE ((Barkai, 2001), (Metzler et al., 1999)[2]). However, in our case, we still run into difficulty since the diffusion coefficient is a location-dependent inhomogeneous diffusion tensor (that is, different dimensions have distinct diffusion coefficients) instead of a single scalar. Luckily, as we will discuss in the next section, we are able to approximate the diffusion tensor as a single scalar function late in training.

### 3.2 Fractal Dimensions and Subdiffusion

#### 3.2.1 Singular Learning Theory and Fractal Dimensions

In order to capture the local geometric structure that impacts the diffusive process we make use of singular learning theory (Watanabe, 2009) via the local learning coefficient (LLC) (Lau et al., 2024). We give a brief introduction to these ideas here, but a more substantial introduction is given in appendix A.

Consider our loss function to be the Kullback-Leibler divergence[3] $\mathcal{K}_m[w]$. Consider then the ball $B_r(w^*)$ of radius $r$ about some "true parameter" $w^*$ such that $\mathcal{K}[w^*] = 0$. Letting $\epsilon$ be some arbitrarily small constant, and denote the set of parameters which have loss $K_m[w] < \epsilon$ within the ball $B_r(w^*)$ of radius $r$ as $B_r(w^*, \epsilon)$. Consider then the *singular integral*

$$V(\epsilon) = \int_{B_r(w^*,\epsilon)} \rho(w)dw \qquad (5)$$

where $\rho(w)$ is some arbitrary choice of prior distribution on the parameter space. Now letting $0 < a < 1$ be some arbitrary constant, the *local learning coefficient* Lau et al. (2024) is defined as

$$\lambda(w^*) = \lim_{\epsilon \to 0} \frac{\log \frac{V(a\epsilon)}{V(\epsilon)}}{\log(a)} \qquad (6)$$

We then have that asymptotically as $\epsilon \to 0$ (under some mild assumptions):

$$V(\epsilon) \propto \epsilon^{\lambda(w^*)} \qquad (7)$$

In the diffusion picture, the LLC behaves as a localized *mass (Minkowski-Bouligand) fractal dimension* which determines the geometry of (potentially degenerate) near critical points. The nature of this relationship is discussed in greater depth in appendix B.1.

### 3.3 Laws of Fractal Diffusion and SGD

#### 3.3.1 The Spectral Dimension

While the LLC captures the geometry of the loss, we all need to capture the dynamics of SGD on this geometry. To this end we utilize a second fractal dimension which describes the trajectory of particles under a potential called the *spectral dimension* $d_s$ ((Millán et al., 2021), (Bouchaud & Georges,

---

[1]Since this is a continuous time formulation, it is more accurate to call it a stochastic gradient flow, however the former is known to be reasonably well-approximated by the latter (Li et al., 2019).

[2]The super-diffusive component matters for studying things like relaxation and crossover time.

[3]We can just as well use the log loss (as it only differs by an additive constant) but using the KL-divergence simplifies the analysis.

1990)). We start with the definition in the "homogeneous" case (e.g when the fractal dimension of the medium is the same everywhere) and then adapt it to our multifractal case. If we consider the LLC as being the scaling exponent for the volume of "good parameters" in a particular area, the spectral dimension determines then the volume of states that the diffusive process over that area can actually reach over some period of time (in our case, the volume of states SGD can actually reach in that area). We define this dimension below.

**Definition 3.1** (Spectral Dimension (Bouchaud & Georges, 1990)). Let $V_s(t)$ be the total volume of occupied states which result from running a diffusive process on porous media from some initial distribution. The spectral dimension $d_s$ is then defined as:

$$V_s(t) \sim t^{\frac{d_s}{2}} \tag{8}$$

In non-homogeneous systems one can have that the spectral dimension changes on different timescales $\{t_1, ..., t_m\}$ where we have different scaling exponents $d_s(t_i)$ such that

$$V_s(t) = t^{\frac{d_s(t_i)}{2}} \tag{9}$$

if $t$ belongs to timescale $t_i$. If the variation between these different dimensions is too large, the theoretical framing becomes more difficult. Luckily, we find that for SGD this spectral dimension is well-captured by a single constant over training. Thus for vanilla SGD we can use what is known as the asymptotic spectral dimension defined as:

$$V_s(t) \sim t^{\frac{d_s^\infty}{2}} \text{ as } t \to \infty \tag{10}$$

and simply take $d_s = d_s^\infty$((ben Avraham & Havlin, 2000), (Paladin & Vulpiani, 1987)). More information and intuition about the spectral dimension is provided in appendix B.2

### 3.3.2 WEIGHT DISPLACEMENT AND FRACTAL DIMENSIONS

We would now like to figure out the relationship between the local fractal dimension (the local learning coefficient) and the spectral dimension.

**Definition 3.2** (Walk Dimension ((Paladin & Vulpiani, 1987), (Bouchaud & Georges, 1990))). Let $R(t)$ be the displacement of a particle at time $t$. The *walk dimension* is defined by

$$R(t) \sim t^{\frac{1}{d_{\text{walk}}}} \tag{11}$$

with $d_{\text{walk}}$ being the *walk dimension* with $d_{\text{walk}} > 2$ for sub-diffusion.

We note here that these values are defined almost identically even when the process displays initially super-diffusive dynamics (details in appendix B.3).

It is known that in particular regimes the walk dimension takes on a particular form. This is known as the *Alexander-Orbach (AO) relation* and relates the walk dimension to the fractal dimension of the medium (the LLC for us) and the spectral dimension. While originally stated in the context of homogeneous media this relation is known to hold locally for porous media which are homogeneous on a sufficiently small scale (Hambly et al., 2002). Restating these results in our framing gives:

**Theorem 3.1.** *The walk dimension at a point $w_t$ on the loss surface can be given as*

$$d_{walk}(t) = \frac{2\lambda(w_t)}{d_s} \tag{12}$$

*near critical points.*

The idea that neural networks trained by SGD are close to some critical point is a direct result of the prevalence degenerate saddle points of the loss surface ((Dauphin et al., 2014), (Advani et al., 2020), (Fukumizu & Amari, 2000), (Choromanska et al., 2015)). In cases where there are no nearby saddle points (which is more common in early training), this does relation does not need to hold as the diffusion is dominated by the gradient behaviour. This is the sense in which the relation is local. However, as noted, so long as this behaviour is largely isolated to early training it does not impact the theoretical results.

### 3.3.3 DIFFUSION COEFFICIENTS AND LOCAL BEHAVIOUR

We would like to use these fractal dimensions to define a diffusion coefficient. Importantly, we find that the diffusion coefficient is reasonably approximated by a scalar function. We also should expect that late in training, the localized dynamics nearby degenerate points should be directly proportional to the volume of low loss parameters. We state the results informally below, with the formal results and proofs in appendix D.

First we state the following theorem:

**Theorem 3.2** (Small Scale Dynamics are LLC Dependent). *Let $w^* \in W$ be a point such that the Hessian of the loss $H(w^*)$ is positive semidefinite. Let $\mathcal{W} = B(r, w^*)$ be a small area about $w^*$. If the diffusion coefficient $D$ along degenerate directions is isotropic then for fixed error tolerance $\varepsilon$ the first passage time $T(\varepsilon)$ through $\mathcal{W}$ is $\propto \frac{\epsilon^{\lambda(w^*)}}{DC}$ where $C$ is the "capacitance" of the escape set.*

An important thing to note about the above theorem is that this is what would be considered a "pore scale" model of the diffusion, and is determined by the small scale dynamics and as such is less useful for experimentally capturing the behaviour. For experimental purposes we develop a more coarse-grained theory which relies on the following result:

**Lemma 3.1** (Diffusion Coefficient Approximation (informal)). *For reasonable choices of the learning rate in the large batch size regime, the diffusion tensor can be approximated by a scalar function for long runtimes.*

Since the steady state is determined by the long-runtime dynamics, we can study the FFPE with a scalar diffusion coefficient.

To study the coarse-grained diffusive behaviour use a physics-inspired scalar diffusion coefficient for porous media that captures the essential behaviors of the diffusion at some choice of measurement scale called the *characteristic length scale $\xi$*. Under some assumptions we have the following:

**Lemma 3.2.** *Letting $\xi$ be some characteristic length scale, the diffusion coefficient can be approximated as $D_\xi = \xi^{2-d_{walk}}$.*

A thing to note is that the choice of $\xi$ is effectively how far we are zooming out and averaging over the local dynamics, which gives a scaling law, not an exact relation. A general practice is to pick a value of $\xi$ which is large enough to average out the fluctuations in an area but not so large that it starts to ignore large scale changes in structure. The effect of choice of $\xi$ is shown in section 4. One may also notice that this implies that the diffusion coefficient is higher for a small LLC which seemingly contradicts the pore-scale diffusion derived earlier where low LLC was slower. However, this is explained through the spectral dimension (as we shall see) as it is bounded above by the LLC, meaning that a small LLC is only faster if the dynamics allow very free movement inside of the domain.

Combining lemma 3.2 with the definition of the walk dimension given earlier we get:

**Corollary 3.1** (Fractal Effective Diffusion Coefficient). *Let $\xi$ be a choice of the characteristic length scale. One can then define the effective (local) diffusion coefficient for length scale $\xi$ as*

$$D_\xi(w) = \xi^{2 - \frac{2\lambda(w_t)}{d_s}} \tag{13}$$

### 3.4 STATIONARY STATES OF THE SGD FOKKER-PLANCK EQUATION

We now present theoretical results about the diffusive process with proofs in appendix D. In appendix D we provide a brief discussion of impacts when particular assumptions about the system are not met and how the results here can be extended.

Assuming some fixed scale $\xi$, using the effective diffusion coefficient, we can actually find the local steady-state solutions for the SGD Fractional Fokker-Planck equation (if it exists):

**Lemma 3.3.** *Consider a subset $\mathcal{W} \subset W$ such that the effective diffusion coefficient $D_\xi$ is (approximately) constant on $\mathcal{W}$. Suppose then that there exists steady-state solutions of the SGD-FFPE on this subset with true parameter(s) $w^*$ so $\mathcal{D}_t^\alpha p(w^*, t) = 0$. The steady-state $p_s(w|X_m) = p_s(w)$ distribution is then given by $p_s(w) \propto e^{\frac{-\gamma \mathcal{L}_m[w]}{D_\xi}}$.*

Note that the above holds even if $D_\xi$ does not have the form given in definition 3.1 so long as it is simply a scalar. However, if it does have the form, due to the definition of $D_\xi$, the above condition that it be constant is actually simply saying that $\lambda(w)$ be locally constant in $\mathcal{W}$ which tends to be the case away from phase transitions (Wang et al., 2025). We can also get from this a relationship with the Bayesian posterior perspective of singular learning theory.

**Corollary 3.2.** *Letting $\gamma = 1$ for simplicity, if $\mathcal{L}$ is the log-loss and $w \in \mathcal{W}$ then*

$$p_s(w)^{mD_\xi} \propto p(X_m|w) \tag{14}$$

*so*

$$p(w|X_m) = \frac{\rho(w)p_s(w)^{mD_\xi}}{Z_{mD_\xi}} \tag{15}$$

*where $Z_{mD_\xi}$ is the partition function and $\rho$ is an arbitrary choice of prior.*

This explains the observed relationships between Bayesian sampling and SGD seen in (Mingard et al., 2020). We can see that SGD effectively scales the likelihood of certain states of the underlying purely Bayesian distribution at the measurement scale $\xi$ based on how accessible they are to the model under the optimization process. That is, the distribution of solutions found by SGD from some initial distribution concentrate more heavily in particular areas than the Bayesian posterior since SGD simply cannot reasonably reach those areas.

Another important aspect of the local learning coefficient is that it can be considered the quantity that "bounds" the movement of network weights. For notational simplicity let $w(t)$ be the parameters of the system at time $t$, so we can formally state the above as:

**Lemma 3.4.** *Suppose the loss function $\mathcal{L}$ is non-convex and non-constant on $W$. Then with spectral dimension $d_s$ as $t \to \infty$ with fractal dimension $\lambda(w(t))$ on $\mathcal{W} \subset W$, the inequality $d_s \leq \lambda(w(t))$ holds (in the small learning rate regime).*

In the above lemma the timescale condition is used to account for the fact that at early times such sub-diffusive processes can appear nearly linear. Given the above, we get the following corollary:

**Corollary 3.3.** *For time $t$ as $t \to \infty$, we have $d_s \leq \bar{\lambda}(w(t))$ where*

$$\bar{\lambda}(w(t)) = \lim_{\tau \to \infty} \frac{1}{\tau} \int_0^\tau \lambda(w(t))dt \tag{16}$$

Notice that since small $\lambda(w)$ implies greater local volume, but larger $d_s$ implies that the volume spreads faster over time, large local volumes trap the spread of SGD, slowing it down. This aligns with previous research examining the eigenvalues of the Hessian of the loss (Sagun et al., 2016). In the next section we will show that the above result holds experimentally as well as examine other properties of our fractal diffusion theory of SGD.

## 4 EXPERIMENTAL RESULTS

### 4.1 DIFFUSIVE BEHAVIOUR

Here we present experimental results to validate the diffusive theory across multiple model architectures and tasks. Namely we look at small language models trained on the TinyStories dataset(Eldan & Li, 2023), vision models trained on Tiny Imagenet (Le & Yang, 2015), as well as extensive ablations on the MNIST dataset (Deng, 2012) with fully connected architectures with ReLU activations and batch normalization. More extensive experimental details can be found in appendix G.

To compute the LLC we utilize the estimator provided by (van Wingerden et al., 2024). To compute the spectral dimension $d_s$ we first compute the value $\log(R(t))$ where $R(t)$ is the total weight displacement at time $t$. We then find $d_s$ by solving the linear regression problem

$$\log(R(t)) = \frac{d_s}{2\lambda(w)} \log(t) + c \tag{17}$$

where $c$ is simply an offset term.

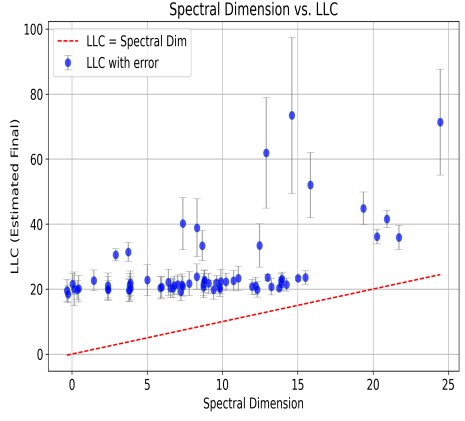



(a) Visualization of lemma 3.4 (MNIST)

(b) Correlation between learning coefficient (average) and total weight displacement (MNIST).

Figure 2: In a) we check that the result of lemma 3.4 holds. In b) we check that independent of our choice of diffusion model, the total displacement and average learning rate are strongly correlated in the large batch, low learning rate regime.

| Model name | $\lambda$ | $d_s$ | $\alpha$ | $r^2$ |
|---|---|---|---|---|
| TinyStories-1M | 32 | 21.422 | 0.33 | 0.98 |
| TinyLlama-15M | 76.1 | 48.3 | 0.32 | 0.98 |
| TinyStories-33M | 39.3 | 38.7 | 0.49 | 0.98 |
| ResNet18 | 72.05 | 0.57 | 0.004 | $\approx 1$ |
| ResNet34 | 73.5 | 0.62 | 0.004 | $\approx 1$ |
| VGG16 | 159.7 | 0.14 | 0.001 | $\approx 1$ |

Table 1: Results for different models.

Using this setup, we are able to experimentally test the result of lemma 3.4 and corollary 3.3, which can be seen in figure 2 for an extensive collection of various models over MNIST, as well as various vision and language models in table 1. We also check the accuracy of the sub-diffusion model.

We find that in general, the sub-diffusive prediction is very accurate for most models tested which are trained to convergence. In particular we note that despite our theory not explicitly accounting for adaptive optimizers and learning rate schedulers, the dynamics vision models fine-tuned using an initial adaptive optimizer, followed by a low learning rate SGD are well-predicted by the theory. Furthermore, by taking pretrained language models which have already been trained to convergence in the weights and then continuing training on their initial pretraining dataset agrees with the predictions of the theory. More results are available in appendices I and H.

### 4.2 POSTERIOR CONCENTRATION

In order to check the results of lemma 3.3 and corollary 3.2 we train a large number of identical fully connected networks on a generated moons dataset (Pedregosa et al., 2011) using SGD. To compare the distribution of solutions found via SGD vs. the (local) Bayesian posterior, we use SGLD (Welling & Teh, 2011) to approximate the Bayesian posterior. We then identify clusters of solutions, and identify the concentrations of SGD and Bayesian solutions within each cluster. To select the scale $\xi$ for tempering, we check how the choice of $\xi$ impacts the KL-divergence between the empirical SGD distribution and the theoretical SGD distribution (figure 3). We can see in figure 4a that the solutions found by SGD do tend to concentrate around lower LLC areas. Figure 4b and table 2 shows how the tempering of the distribution of SGD solutions effectively agrees with approximate Bayesian posterior of SGLD.

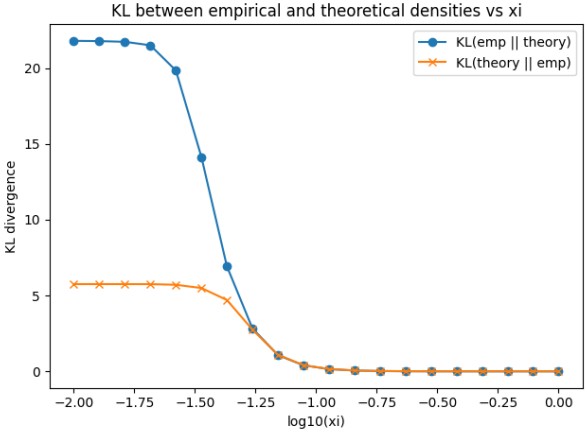

Figure 3: KL-divergences between the empirical vs. theoretical distribution for different choices of $\xi$.

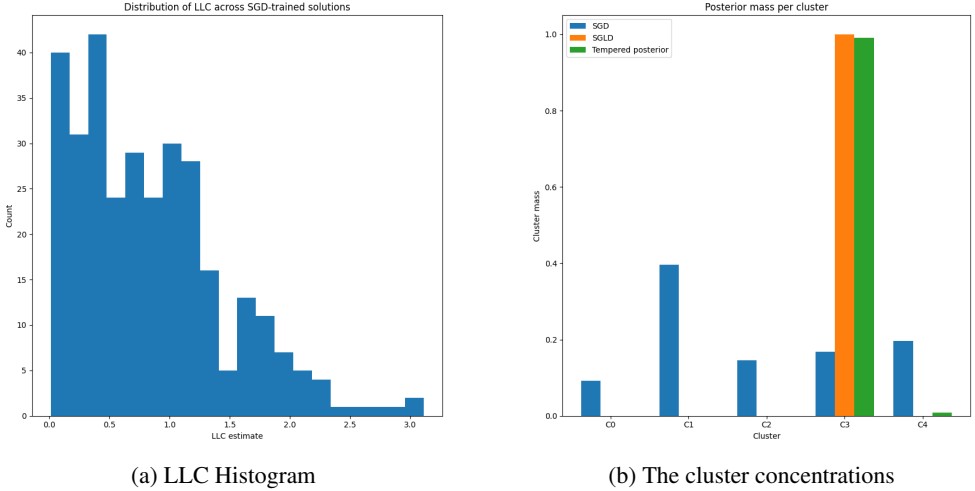

(a) LLC Histogram                    (b) The cluster concentrations

Figure 4: a) Shows the histogram of local learning coefficients of solutions found by SGD. Notice that as predicted by the theoretical results, they tend to concentrate near lower LLC values (better generalizing solutions). b) The probability concentrations of solutions found by SGD (blue), the approximate Bayesian posterior (orange), and the tempered SGD distribution (green) for each cluster. Notice that despite SGD itself preferring the cluster C1, after tempering ($\xi = 0.5$), the tempered SGD steady state distribution almost entirely agrees with the Bayesian posterior. Statistical measures can be seen in table 2.

## 5 DISCUSSION

### 5.1 LIMITATIONS

While we believe our theory is useful and tends to capture dynamics of most optimizers empirically, it does not explicitly take into account complex dynamics of adaptive optimizers like Adam (Kingma & Ba, 2017) as adaptive optimizers can exhibit multiple spectral dimensions over the course of training meaning the theory here is incomplete and should occur as a "special case" of a more general theory. Some experimental results and discussions around this can be seen in appendices I and H.

Another limitation to consider is that we assume the existence of an approximate steady state. While this is a common practice in the study of SGD ((Pesme et al., 2020), (Mandt et al., 2016a), (Mandt et al., 2018)). In general, SGD iterates do not converge to exact equilibria, but under standard assumptions and suitable learning-rate schedules (or a sufficiently small learning rate) they approach

| Metric | Value |
|---|---|
| $\mathcal{K}$(Bayes$\|$Tempered SGD) | 0.009 |
| Wass(Bayes, Tempered SGD) | 0.002 |
| JS(Bayes, Tempered SGD) | 0.003 |

Table 2: The KL divergence, the Wasserstein distance, and the Jensen-Shannon divergence for the approximated Bayesian posterior and the tempered SGD distribution.

the set of stationary points and attain iterates with small gradient norm. One can in theory have instances where no approximate steady state exists since label noise can in theory produce a non-equilibrium flow through states, so SGD might have a non-equilibrium steady state with probability flow driven by said noise. While examining such situations is outside the scope of this work, it is an important avenue of future work to examine a) the time to equilibrate of SGD and b) if it does not equilibrate, can we quantify its non-equilibrium steady state?

## 5.2 CONCLUSION AND AVENUES FOR FUTURE WORK

Here we have argued that the long runtime dynamics of SGD are captured by taking the corresponding Fokker-Planck equation to describe diffusion on a porous geometry. This porous geometry corresponds is described by the learning coefficient, drawing a direct relationship between the dynamics of SGD to Bayesian statistics via singular learning theory. Our experimental results validate this claim.

We believe our theory helps provide insight into the learning process and adds to the groundwork needed to build a foundational theory of learning dynamics. Our theory says that the learning process is governed partially by the model's behavioral phases as described by the learning coefficient. This opens up a framework for studying emergence and phase transitions during training by considering properties of the dynamical system. Adapting this framework explicitly to adaptive optimizers and checking how this impacts the diffusive structure is an important avenue for future work.

### 5.2.1 REPRODUCIBILITY STATEMENT

To encourage reproducibility we provide source code for the experiments included along with extensive documentation in appendices J and I.

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

## A  SINGULAR LEARNING THEORY BASICS

Here we give an informal introduction to singular learning theory. For more in-depth but still accessible introduction, we recommend the *Distilling Singular Learning Theory* series of blog posts (Carroll, 2023), along with the the seminal work by Watanabe (Watanabe, 2009). For us, it's mostly important to understand the problems that singular learning theory solves. To do this, we first must consider a classical idea in machine learning, the *Bayesian Information Criterion*. The BIC is used to determine which model from a set of different models is likely to generalize the best. Let $a_w$ be a model with $d$ free parameters $w$ in the collection of models, trained over $m$ datapoints and denote the minimum loss achievable by $a_w$ as $L_n(a_w^0)$. The BIC says that we should select the model from our collection of models which minimizes the following:

$$\text{BIC} := nL_n(a_w^0) + \frac{d}{2}\log n \tag{18}$$

This more-or-less says that we should choose the simplest model that fits our data.

The caveat about the BIC however is it makes the assumption that the models we care about are "regular statistical models". There are two key things that are required for a statistical model to be

regular. First, the model must be *identifiable*, which effectively means that any set of parameters for $a$ are unique in that if $a_{w_1}(x) = a_{w_2}(x)$ then $w_1 = w_2$. Second, the Fisher Information matrix near the true parameters $a_w^0$ must be positive definite. This condition is easiest to understand if we assume the loss is the KL-divergence (or log loss), as it corresponds to saying that the Hessian of the loss $H(L_n(a_w^0))$ is non-degenerate, having only non-zero eigenvalues.

This fact is key for how one derives the BIC. While the formal derivation of the BIC is straightforward, it is time consuming and there's a much simpler way to intuitively see why it matters. First, the non-degeneracy of $H(L_n(a_w^0))$ means the geometry of the loss surface is a paraboloid about $a_w^0$. We want to then measure how many configurations of $a$ have a loss less than $\epsilon$, so we want to measure the volume of a paraboloid of height $\epsilon$ in our parameter space. The nice thing about a parabaloid is that its volume is half of that of the cylinder that encloses it. This can be computed straightforwardly from the $d$-dimensional volume of tubes formula(Weyl, 1939):

$$\frac{V_d(2\epsilon)^{\frac{d}{2}}}{\sqrt{\det(H(L_m(a_w^0)))}} \tag{19}$$

Here $V_d$ is the volume of the $d$-sphere. This formula is effectively where the $\frac{d}{2}$ comes from in the BIC. Now, one might notice that if we are considering potentially degenerate local minima, this formula cannot be applied since the degeneracy of the Hessian means the determinant is 0. In this case, the BIC is not well defined either. Models with degenerate minima are called singular models. In some sense, most model classes are singular. Neural networks for instance are highly singular and generally admit many equivalent parametrizations for computing the same function. Singular learning theory attempts to handle this problem by finding a method for computing the volume of degenerate local minima.

This is done by considering the *singular integral*(Watanabe, 2009):

$$V(\epsilon) = \int_{\{w \in W | L(w) < \epsilon\}} \rho(w) dw \tag{20}$$

where $\rho(w)$ is a prior distribution over the parameter space so $\rho(w)dw$ behaves as a measure, and $L$ is the population loss. Unlike the quadratic minima case, there is no straightforward volume formula one can use here. However, as is shown in (Watanabe, 2009), as $\epsilon \to 0$ asymptotically this integral is:

$$V(\epsilon) = c_1 \epsilon^\lambda (-\log \epsilon)^{m-1} + o(\epsilon^\lambda (-\log \epsilon)^{m-1}) \tag{21}$$

where $\lambda$ is *learning coefficient* and $m$ the *multiplicity*. One can define this integral at level sets which are non-true parameters by shifting the value in the integrand like $\{w \in W | 0 < L(w) - \delta < \epsilon\}$.

In short, one can show that the volume about a local minima scales with the height $\epsilon$ according to the learning coefficient $\lambda$ so the volume of the degenerate minima scales $\propto \epsilon^\lambda$ as $\epsilon \to 0$ (Watanabe, 2009). This can be used to derive the *Widely Applicable Bayesian Information Criterion* (Watanabe, 2012) which is given by:

$$\text{WBIC} := nL_n(a_w^0) + \lambda \log n \tag{22}$$

The natural interpretation of $\lambda$ is as the "effective dimension" of a model. We note here as well that it is relatively common to treat the multiplicity as taking the value $m = 1$ to simplify working with singular models as the relative contributions for most applications are negligible in their effects.

## A.1 THE LOCAL LEARNING COEFFICIENT

In the above, we discussed the global learning coefficient of (Watanabe, 2009). A local version of this was defined in (Lau et al., 2024). This is given straightforwardly by simply restricting from the singular integral over the whole set of $\epsilon$-true parameters to some local neighborhood of some parameter of interest $w^*$. One normally considers a ball of some radius $r$ about said parameter $B_r(w^*)$ and then defines the local singular integral as

$$V_{w^*}(\epsilon) = \int_{\{w \in B_r(w^*) | L(w) < \epsilon\}} d\mu(w) \tag{23}$$

where $d\mu(w)$ is the standard Lebesgue measure. The *local learning coefficient* and *local multiplicity* are of the same form as the global case:

$$V_{w^*}(\epsilon) \approx \epsilon^{\lambda(w^*)}(-\log \epsilon)^{m(w^*)-1} \tag{24}$$

When looking at the local learning coefficient, one must make a choice of scale $r$. However, we note that increasing $r$ cannot increase the value of $\lambda(w^*)$. This comes from a fundamental property of the learning coefficient which says that the learning coefficient in an area is the smallest of all possible local learning coefficients as $r \to 0$. The reasoning for this is non-trivial, and the interested reader is referred to (Watanabe, 2009) for details.

We now briefly explain how the local learning coefficient is computed in practice. Suppose that we define some distribution over $B_r(w^*)$. The local learning coefficient estimator of (Lau et al., 2024) is given as

$$\hat{\lambda}(w^*) = \frac{n}{\log n}[\mathbb{E}_{w|B_r(w^*)}(L_n(w)) - L_n(w^*)] \tag{25}$$

This accords with our intuition that if $w^*$ is simple/flat then perturbing the value of $w^*$ should not change the loss. For in-depth experimental results for the accuracy of this estimator we refer readers to (Lau et al., 2024) and (Wang et al., 2024b).

### A.1.1 Dependence of the LLC on $r$

While this question is addressed more formally in appendix B of (Lau et al., 2024), we address this point here informally. Under the standard assumptions of Singular Learning Theory (Watanabe, 2009), the local learning coefficient on a ball of radius $r$ around a true parameter $w^*$ converges to some fixed "pointwise" learning coefficient. This is because as $r$ becomes small, the behaviour of the loss $L(w)$ for some $w \in B_r(w^*)$ is dominated by the leading order of the singular expansion about $w^*$. Shrinking $r$ cannot change the monomial exponents obtained after resolution of singularities; those exponents are geometric invariants of the germ of $L$ at the point.

## B Fokker-Planck Equations and Fractal Dimensions

### B.1 The Mass Dimension

In section 3.2.1 we claim that the local learning coefficient is effectively a mass dimension. To see this, let's start by considering a large collection of (non-interacting) particles diffusing through an arbitrary fractal media. An important thing to note here is that diffusion on fractal media is actually a special case of the more general "diffusion on porous media" where the volume of the pores scales like a fractal dimension. Keeping notational consistency, we are interested in the valid states in some ball $B(w^*)$. To measure this, we need something called the "characteristic linear dimension" which we can scale asymptotically. In porous media, this is something like the "pore diameter" (since particles can occupy any point in a pore). For consistency again, we denote this value as $\epsilon$.

The mass dimension is then the fractal dimension that determines the relative volume of the pores to the total volume as we restrict the diameter of the pores by taking $\epsilon \to 0$. One way to see what this is doing is to consider the mass dimension of an empty sphere (that is, the whole thing is a pore and nothing is there to impede a particle). As we take $\epsilon$ to 0, we end up with every possible point being a pore, so the relative volume is 1.

So we can imagine that for some ball centered about a reference point $B(w^*)$, and we care about the volume of states a particle could exist in within this ball, usually denoted as $M(\epsilon)$. The relative volume is given identically to the learning coefficient case like

$$\frac{M(\epsilon)}{M(B(w^*))} \tag{26}$$

We get the fractal dimension which determines the relative volume $d_f(w^*)$ as:

$$M(\epsilon) \propto \epsilon^{d_f(w^*)} \tag{27}$$

as $\epsilon \to 0$ asymptotically. One can see that this coincides with the definition of the local learning coefficient((Kinsner, 2005), (Bouchaud & Georges, 1990)). From a fractal geometric viewpoint, the

normal mass dimension is computed as a "Minkowski sausage" which can be thought of as similar how the volume of a "sausage casing" wrapping a pore scales as you decrease the radius. The learning coefficient is similar, except it uses a different "gauge function" since our pores are not tubes, but are instead like basins, so we look at how the volume of water in the basin changes as we decrease the height of the water. This fractal dimension has been used to model the diffusion of water through ground soil of different types (Tyler & Wheatcraft, 1990). They find diffusion is much slower and has a larger fractal dimension through clay-like media where there are very few channels to move through, where more sand-like media has faster diffusion and a lower fractal dimension. This is conceptually the same as the results given here.

## B.2 THE SPECTRAL DIMENSION

The spectral dimension is easiest to understand if you think "how fast does diffusion manage to explore new places?" rather than "what is the geometric dimension of the space?". In the main text we introduce the spectral dimension as the scaling exponent of the "volume of visited/occupied states". That is, $V_s(t)$ simply denotes the volume of states which have been visited by the process at time $t$. The spectral dimension is then:

$$V_s(t) \sim t^{\frac{d_s}{2}} \tag{28}$$

Intuitively $d_s$ says how fast diffusion fills out the medium you are diffusing on, and in that sense it is the "dimension that diffusion sees".

In this work, the medium is (a region of) parameter space, and the diffusive process is the long-run stochastic motion of SGD (modeled via the Fokker–Planck equation). The spectral dimension then measures how quickly SGD can spread over the set of weight configurations that are dynamically accessible from some initial condition. To see this a bit more formally we will discuss the spectral dimension of normal Brownian motion.

### B.2.1 SPECTRAL VS. GEOMETRIC DIMENSION

On a flat surface like $\mathbb{R}^d$, a random walk driven by Brownian motion has a displacement $R(t) \propto t^{\frac{1}{2}}$ so the region which gets explored after time $t$ can be seen straightforwardly to have a volume which simply scales with the displacement, since the diffusion is uniform in $d$ dimensions. That is we have:

$$V_s(t) \propto R(t)^d \tag{29}$$

so

$$V_s(t) \propto t^{\frac{d}{2}} \tag{30}$$

One can see though that by introducing obstructions into this free space changes the diffusion rate. Moving through a medium with many obstructions changes the geometry that diffusion experiences, which can be very different from the naive ambient dimension. Narrow channels, dead ends, bottlenecks, and local degeneracies all slow down or redirect the random walk. The process might live in a very high-dimensional ambient space, but only a much smaller effective set of directions is actually accessible on the timescales we care about. In such cases one typically has that $d_s \neq d$. In our setting, the ambient dimension is the number of parameters, while the local "mass dimension" of low-loss regions is given by the local learning coefficient $\lambda(w)$. Consider this in the same context as the Brownian motion example. Since the learning coefficient captures the volume of low loss states, a diffusive process on those states can only ever access at most that many states, bounding the spectral dimension. So the spectral dimension literally captures the number of states SGD can actually visit over some time frame.

### B.2.2 WHY SGD HAS A SPECTRAL DIMENSION

For simplicity we are going to imagine a very localized picture of SGD here. Suppose we initialize in some area $\mathcal{W}$ which has learning coefficient $\lambda$ which approximates the volume of low loss points in $\mathcal{W}$. The spectral dimension $d_s$ of this area tells us how efficiently the SGD-induced diffusion spreads into that volume over time. The walk dimension $d_w = \frac{2\lambda}{d_s}$ ties the two together and governs the displacement scaling.

From the perspective of diffusion theory, $d_s$ is therefore the right quantity to describe the effective dimensionality of SGD dynamics.. It is "spectral" because, in principle, it could be read off from

the spectrum of the Fokker–Planck operator governing SGD; in practice, we estimate it through the observed power-law scaling of displacement, which is equivalent information in the regime we study.

### B.3 Impact of Early super-diffusive Dynamics on Diffusion Exponents

Given the displacement equation

$$R(t) \sim t^{\frac{1}{d_{\text{walk}}(t)}} \tag{31}$$

we can give a rather straightforward way to incorporate the super-diffusive component. If a given trajectory goes from super-diffusive to sub-diffusive, there exists a crossover time $t_c$ where the dynamics change. Given this, let $r(t)$ be the super-diffusive component such that for $t > t_c$ we have $r(t) = r(t_c)$. Letting $I_{t_c}(t) = 1$ for $t \geq t_c$ we can then write the displacement as

$$R(t) \sim I_{t_c}(t)(t - t_c)^{\frac{1}{d_{\text{walk}}(t)}} + r(t) \tag{32}$$

A similar trick works for the volume. This means that one can account for the early super-diffusive behaviour without directly impacting the exponents if one accounts for the crossover time. In general, we find experimentally in many cases that the initial super-diffusive regime is short enough that the dynamics are still well approximated by the entirely sub-diffusive equations.

## C   Towards Practical Applications

While the results here are largely theoretical, we believe they provide important avenues and insights for practical applications. We discuss some of these below.

### C.1   Transfer Learning And Robustness

#### C.1.1   Parameter Choice for Transfer Learning

In transfer learning, you start from a pretrained minimum and fine-tune with SGD on a new task. The value of $\lambda$ at initialization tells you how wide that basin is. If one maintains a record of the weight displacement from pretraining, one can estimate how the effect the new data distribution has on $d_s$ during the initial steps. These tell you how "wide" and "connected" the accessible region is, which can inform how aggressively to tune the learning rate and batch size. For example if the spectral dimension is low, but the loss is high, you are likely stuck in a wide flat basin, so one might increase the learning rate or decrease the batch size.

#### C.1.2   Robust Model Selection

Our theory indicates that model parameters with a low $\lambda$ but a high relative spectral dimension $d_s$ represent models which had more movement within the same large basin. Selecting for such models might result in more robust generalizing models as the minima they exist in are "flat"

### C.2   Learning Rate Schedulers and Optimizers

#### C.2.1   Designing Learning Rate Schedulers

Warmup and decay can be viewed as shaping $d_s$ over time. This suggests the potential for structural schedule design: e.g. maintain higher $d_s$ early (more exploration), then lower $d_s$ later (stronger localization).

#### C.2.2   Evaluating Optimizers

Another application is evaluating optimizers for particular structural properties. That is, one can look at how the spectral dimension or the learning coefficient change over time and compare these with SGD to better understand how the optimizer impacts generalization behaviour.

### C.3 APPROXIMATE BAYESIAN INFERENCE

One can potentially apply the theory here to calibrating uncertainty in SGD. In practice, "Bayesian" approximations often assume Langevin dynamics with quadratic minima. Our theory gives a way to correct for degeneracy and accessibility so that posterior variances and predictive intervals reflect the actual dynamics of SGD, not an idealized model.

## D PROOFS

### D.1 LLC DETERMINES PORE SCALE DYNAMICS NEAR DEGENERATE POINTS

The proof of this relies on results from the theory of first passage times, homogenization, and porous diffusion, namely the estimation of mean first passage times of porous media. However, the components needed are straightforward to state. First we need the following definition(Baxter, 1985):

**Definition D.1.** Let $K \subset \mathbb{R}^d$. The Newtonian capacity $\text{Cap}(K)$ is defined as

$$\inf\{\int_{\mathbb{R}^d} |\nabla u|^2 dx : u \in C_c^\infty, u \geq 1 \text{ on } K\} \tag{33}$$

Intuitively: if $K$ is "big" or "accessible," you can spread charge thinly and keep fields weak which makes them low energy, large capacity; if it's tiny or shielded, fields must be intense $\rightarrow$ high energy, small capacity. Next we give a known result of first passage times in porous media(Redner, 2001).

**Theorem D.1.** *Let $\Omega \subset \mathbb{R}^d$ be a porous domain and let $dX_t$ be a brownian process on $\Omega$ and let $\Gamma$ be the set of all absorbing walls in $\Omega$. If we consider then $\Gamma_\delta$ to be the subset of absorbing walls with area less than $\delta$ then the mean first passage time of a Brownian particle through $\Omega$ is (asymptotically) proportional to*

$$\bar{T} \propto \frac{|\Omega|}{DCap(\Gamma_\delta)} \tag{34}$$

*as $\delta \rightarrow 0$ where $D$ is the diffusion coefficient of the Brownian process.*

Intuitively this says that for isotropic noise the time spent in some domain is proportional to how big the domain is and how large the escape walls are. We can use this to get the mean first passage time for SGD around degenerate saddle points under some simplifying assumptions.

**Theorem D.2.** *Let $w^* \in W$ be a point such that the Hessian of the loss $H(w^*)$ is positive semidefinite. Let $\mathcal{W} = B(r, w^*)$ be a small area about $w^*$ with local learning coefficient $\lambda(w^*)$ with $\mathcal{W}_\varepsilon = \{w \in \mathcal{W} | \mathcal{L}(w) < \varepsilon\}$. assume we have small istropic noise $D$ and that there is a reflective boundary at height $\varepsilon$ along the wall, and some escape set (absorbing boundary) $\Gamma(\mathcal{W})$ then the time it takes to traverse distance with error tolerance $\varepsilon$ is inversely proportional to the LLC.*

*Proof.* First note that in the above picture, we effectively have Brownian motion along a submanifold $\mathcal{W}_\varepsilon = \{w \in \mathcal{W} | \mathcal{L}(w) < \varepsilon\}$, which we can treat as diffusion through a porous media where the pores have height $\varepsilon$. Then from theorem D.1 we know that asymptotically as $\delta \rightarrow 0$ we should have

$$\bar{T}(\mathcal{W}_\varepsilon) \propto \frac{|\mathcal{W}_\varepsilon|}{D\text{Cap}(\Gamma_\delta(\mathcal{W}_\varepsilon))} \tag{35}$$

and since we can take $|\mathcal{W}_\varepsilon| = V(\varepsilon)$. This gives the desired result. $\square$

While the above result requires isotropic noise and zero gradient, analogous results likely hold under weak anisotropy and weak gradients by considering potential driven Brownian motion on a submanifold which is tilted to behave like a porous media.

### D.2 DIFFUSION COEFFICIENT HAS A SCALAR APPROXIMATION

First we would like to prove that the diffusion coefficient is well-approximated by a constant. To do this we prove a handful of results. In the following let $\gamma \ll 1$ be a small learning rate and let $n$ denote the batch size. Letting $\mathcal{D}_t^\alpha$ be the Caputo fractional derivative, take

$$\mathcal{D}_t^\alpha w_t = -\gamma \nabla \mathcal{L}_n(w_{t-1}) + \Sigma_{w_{t-1}} \tag{36}$$

to be the overdamped Langevin equation with

$$\Sigma_{w_{t-1}} = \sqrt{2D(w_{t-1})}dW_t \tag{37}$$

where $dW_t$ is a $d$-dimensional Wiener process. Let $T$ be the time to equilibration for an instance of the system. Then as $t \to T$, assume for almost all eigenvalues $e_i$ of the Hessian $H(w_{t-1})$ of $\mathcal{L}_n(w_{t-1})$ we have $e_i \ll 1$ are $\approx 0$ (which has been shown in (Sagun et al., 2016)). Furthermore, we start with the assumption that the full diffusion tensor is proportional to the Hessian so $D(w_{t-1}) \propto H(w_{t-1})$ which has been shown previously (both experimentally and rigorously under particular assumptions, see (Xie et al., 2021) and (Smith & Le, 2018)). We start with the following:

**Lemma D.1** (Effective Diffusion Tensor is Low Rank). *As $t \to T$, assuming $\mathcal{L}$ is in $C^2$, the diffusion tensor is well-approximated by an effective diffusion tensor $D_{eff}(w)$ with rank $d_{eff} \ll d$ with $w \in \mathbb{R}^d$.*

*Proof.* Since $D(w_{t-1}) \propto H(w_{t-1})$ and we know that almost all eigenvalues are $\approx 0$, the result follows almost immediately from the Eckart–Young–Mirsky theorem. That is, approximating via the truncated eigendecomposition

$$D_k = \sum_i^k e_i q_i q_i^T \tag{38}$$

and considering the ordering $|e_1| \geq |e_2| \geq ... |e_n|$ we get the approximation error

$$\|D - D_k\| = \sum_{i>k} e_i^2 \tag{39}$$

and if $e_i \approx 0$ for all $i > k$, then $\sum_{i>k} e_i^2 \approx 0$. $\square$

**Lemma D.2.** *As $t \to T$, taking $D(w_t) \approx \frac{\gamma}{n}H(w_t)$ (Xie et al., 2021) for batch size $n$ and learning rate $\gamma$. For any $\epsilon$ there exists some choice of $\gamma$ and $n$ such that there is a scalar value $a$ with*

$$\|D(w_t) - aI\| < \epsilon \tag{40}$$

*Proof.* Since $D(w_t)$ is symmetric, it can be rewritten as $Q\Lambda Q^T = D(w_t)$ where $\Lambda = \text{diag}(e_1, ..., e_n)$. We can then take

$$D(w_t) - aI = Q(\Lambda - aI)Q^T \tag{41}$$

and by the unitary invariance of the Frobenius norm we get:

$$\|D(w_t) - aI\| = \|(\Lambda - aI)\| \tag{42}$$

which is

$$\sum_i^d (e_i - a)^2 \tag{43}$$

Then since for almost all $i$, $e_i = 0$ we have

$$= ca^2 + \sum_j (e_j - a)^2 \tag{44}$$

where the sum is over all non-zero eigenvalues and $c$ is the number of 0 eigenvalues. Let $a^* = \text{argmin}_{a \in \mathbb{R}} ca^2 + \sum_j (e_j - a)^2$. Notice that since $e_j$ is an eigenvalue of $\frac{\gamma}{n}H(w_t)$ we can rewrite it $e_j = \frac{\gamma}{n}e_j'$ where $e_j'$ is the corresponding eigenvalue in the unscaled Hessian.

Letting $e_1'$ be the largest unscaled eigenvalue, notice that as $\gamma \to 0$ and/or $n \to \infty$ that the value for $e_1$ dominates the sum, and all the other $e_j$ go to 0, so the sum is then

$$\approx (d-1)a^2 + (e_1 - a)^2 \tag{45}$$

so setting $a = e_1$ we get

$$\|(\Lambda - aI)\| \approx (d-1)e_1^2 \tag{46}$$

$$= (d-1)(\frac{\gamma}{n}e_j')^2 \tag{47}$$

and since the learning rate and the batch size can be made arbitrarily small/large, our result follows. $\square$

We now prove the general form of the scalar diffusion coefficient at some effective scale. This is a well-known result within the diffusion literature (Bouchaud & Georges, 1990) but we include it here for completeness. We prove it for the homogeneous case. The inhomogeneous case follows from application of this to a restricted sub-domain.

**Lemma D.3.** *Let $\xi$ be some choice of length scale and $d_{walk}$ be the walk dimension. The diffusion coefficient can be approximated by a scalar as $D_\xi = \xi^{2-d_{walk}}$.*

*Proof.* The effective diffusivity is defined as $D_\xi = \frac{\text{length}^2}{\text{length traversal time}}$. Since $R(t) \sim t^{\frac{1}{d_{\text{walk}}}}$ we get that $R(t)^{d_{\text{walk}}} \sim t$ so setting $R(t) = \xi$ and rewriting $t(\xi)$ as the time $t$ such that $R(t) = \xi$, we get $t \sim \xi^{d_{\text{walk}}}$ we can write

$$D_\xi = \frac{\xi^2}{\xi^{d_{\text{walk}}}} \tag{48}$$

$$= \xi^{2-d_{\text{walk}}} \tag{49}$$

as desired. $\square$

## D.3 STEADY STATES

Here we will give proofs of the results given in section 3.

**Lemma.** *Consider a subset of the parameter space $\mathcal{W} \subset W$ such that the effective diffusion coefficient $D_\xi$ is (approximately) constant on $\mathcal{W}$. Suppose then that there exists steady state solutions on this subset $w^*$ so $\frac{\partial p(w^*, t)}{\partial t} = 0$. The steady-state distribution is then given by $p_s(w) \propto e^{\frac{-\gamma \mathcal{L}_m[w]}{D_\xi}}$.*

*Proof.* First, by definition of the steady state we have $\mathcal{D}_t^\alpha p(w, t) = 0$ which reduces the fractional FPE to effectively the normal FPE, so we must solve the following PDE:

$$0 = \nabla \cdot (D(w, t)\nabla p(w, t) - \gamma p(w, t)\nabla \mathcal{L}_m[w]) \tag{50}$$

Now under the assumption that for all $w_1, w_2 \in \mathcal{W}$ that $D(w_1) \approx D(w_2)$, then the long-term behavior of the diffusion coefficient at length scale $\xi$ can be approximated by the effective diffusion coefficient given in definition 3.1, giving

$$0 = \nabla \cdot (D_\xi \nabla p(w, t) - \gamma p(w, t)\nabla \mathcal{L}_m[w]) \tag{51}$$

One can also see that the values of $D_\xi$ and $\mathcal{L}_m[w]$ are not dependent on $p$ (that is, the change in the probability of $w$ does not change the loss or geometric properties determining diffusion at $w$) meaning that the SGD-FFPE reduces to a linear partial differential at steady state solutions. The solution is then readily obtained by solving the normal Fokker-Planck equation, which is simply the Boltzmann distribution for the system giving $p_s(w) \propto e^{\frac{-\gamma \mathcal{L}_m[w]}{D_\xi}}$ as desired. $\square$

**Corollary.** *Letting $\gamma = 1$ for simplicity, if $\mathcal{L}$ is the log-loss, then*

$$p_s(w)^{mD_\xi} \propto p(X_m|w) \tag{52}$$

*so*

$$p(w|X_m) = \frac{\rho(w)p_s(w)^{mD_\xi}}{Z_{mD_\xi}} \tag{53}$$

*where $Z^{mD_\xi}$ is the partition function. and $\rho$ is the prior.*

*Proof.* First, note that the empirical negative log loss is

$$\mathcal{L}_m[w] = -\frac{1}{m}\sum_{i=1}^{m} p(y_i|x_i, w) \tag{54}$$

This is a dimensionless quantity, however, we can consider the coarse-graining of the parameter space by some scale $\xi$ so that $w \mapsto B(w, \xi)$. By taking the appropriate choice of measurement scale $\xi$

(given some general regularity assumptions about the structure of the loss surface implicit in singular learning theory) we have that if $w_1, w_2 \in B(w, \xi)$ then $\mathcal{L}_m[w_1] \approx \mathcal{L}_m[w_2]$. Now consider that:

$$e^{-m\mathcal{L}_m[w]} = \prod_{i=1}^{m} p(y_i|x_i, w) \tag{55}$$

$$= p(X_m|w) \tag{56}$$

Now given the result of lemma 3.3 one gets $p_s(w) = \dfrac{e^{\frac{-\mathcal{L}_m[w]}{D_\xi}}}{Z_s}$ for partition function $Z_s$. We then get that

$$(e^{\frac{-\mathcal{L}_m[w]}{D_\xi}})^{mD_\xi} = e^{-m\mathcal{L}_m[w]} \tag{57}$$

$$= p(X_m|w) \tag{58}$$

Letting $Z_{mD_\xi}$ be the appropriate partition function, the result then follows from application of Bayes' theorem. $\qquad\square$

**Lemma.** *Suppose the loss function $\mathcal{L}$ is non-convex and non-constant on $W$. Then with spectral dimension $d_s$ as $t \to \infty$ with fractal dimension $\lambda(w(t))$ on $\mathcal{W} \subset W$, the inequality $d_s \leq \lambda(w(t))$ holds (in the small learning rate regime).*

*Proof.* Consider two points $w_1, w_2$ be two points visited in the long timescale regime at times $t_1, t_2$ separated by distance $R$. If we suppose that there exists a linear path connecting $w_1$ to $w_2$ along the manifold and we remove all other paths linking the two points we have diffusion on a linear structure. Now using the definition of the walk dimension, following the arc $A$ of this restricted structure gives $R_A(t) \propto t^{\frac{1}{d_{\text{walk}}}}$ but since this restricted structure has only a single path, it has walk dimension $d_{\text{walk}} = 2$. Now, suppose that this is true for any pair of points. Notice that this implies that all points are connected by a linear path at arbitrary distances along the loss manifold meaning the loss surface would have $d_{\text{walk}} = 2$. Furthermore this would imply that the loss does not change for any choice of parameter, violating the fact that it is non-constant so we must have $d_{\text{walk}} > 2$. Now since $d_w = \frac{2\lambda(w)}{d_s}$ we have $d_s = \frac{2\lambda(w)}{d_{\text{walk}}}$ and clearly if $d_{\text{walk}} > 2$ then $\frac{2\lambda(w)}{d_{\text{walk}}} \leq \lambda(w)$ so $d_s \leq \lambda(w)$ $\qquad\square$

**Corollary.** *For time $t$ as $t \to \infty$, we have $d_s \leq \bar{\lambda}(w(t))$ where*

$$\bar{\lambda}(w(t)) = \lim_{\tau \to \infty} \frac{1}{\tau} \int_0^\tau \lambda(w(t))dt \tag{59}$$

*Proof.* Let $\tau_0$ be the time such that for all $\tau > \tau_0$, the inequality of lemma 3.4 holds. Consider then a time $T \gg \tau_0$ and consider the integral

$$\int_0^T \lambda(w(t))dt = \int_0^{\tau_0} \lambda(w(t))dt + \int_{\tau_0}^T \lambda(w(t))dt \tag{60}$$

and since $\tau_0$ is finite we can take the first portion of this integral to be a constant (since we know that the LLC is bounded above by $\frac{d}{2}$ where $d$ is the number of free parameters):

$$\int_0^{\tau_0} \lambda(w(t))dt = C \tag{61}$$

By the result of lemma 3.4 we have that for all times greater than $\tau_0$, we must have

$$\int_{\tau_0}^T \lambda(w(t))dt \geq \int_{\tau_0}^T d_s dt \tag{62}$$

and since $d_s$ is constant

$$\int_{\tau_0}^T \lambda(w(t))dt \geq (T - \tau_0)d_s \tag{63}$$

which means that by adding $C$ to both sides and dividing by $T$ we get

$$\frac{1}{T} \int_0^T \lambda(w(t))dt \geq \frac{(T - \tau_0)d_s}{T} + \frac{C}{T} \tag{64}$$

From this we get

$$\frac{1}{T} \int_0^T \lambda(w(t))dt \geq d_s + \frac{(C - \tau_0)d_s}{T} \tag{65}$$

where the term $\frac{(C-\tau_0)d_s}{T}$ vanishes as $T \to \infty$ since $C$ must be finite. $\qquad\square$

## D.4 EXTENDED RESULTS

Below we give an extra result which explains a domain where the theory can fail, as well as a discussion about complexities regarding how the diffusion process relaxes towards stationary states.

In the following result, consider the large-batch small learning rate regime. What happens if instead of the long runtime being sub-diffusive it has a small linear component?

**Proposition D.1.** *Suppose that* $R(t) \sim t^{\frac{1}{d_{walk}}} + ct$ *for small constant and has no stationary distribution.*

*Proof.* The proof is straightforward. Let $\tau_2$ be the crossover time where for all $t > \tau_2$, $ct > t^{\frac{1}{d_{\text{walk}}}}$ so on long timescales the small linear term dominates the sub-diffusive term, so $R(t)$ for $t > \tau_2$ can be effectively approximated as $R(t) \sim ct + R(\tau_2) + o(t^{\frac{1}{d_{\text{walk}}}})$ and as $t \to \infty$ the constant term $ct$ dominates. However, a stationary distribution must be independent of time. This cannot be the case however as at any point in time the diffusive process has non-trivial movement away from initialization so the distribution $p(w, t)$ spreads continuously over all timescales. $\square$

The thing that causes the problem with linear diffusion is if the space is unbounded. If one bounds the space with a reflective boundary one can recover a stationary state but the dynamics become more complicated. One way to approach studying this system would be to assume that the process eventually reaches a global minima and that such minima form a connected submanifold. One could then consider certain directions on the manifold to be confining, and others to be free. Processes on this submanifold can be studied using tools like Morse-Bott theory.

## E HOMOGENIZATION

Ultimately the theory presented here relies on the process of homogenization, which is a well-known technique in the study of diffusion. We will give a basic informal overview here, but a full treatment can be found in (Cioranescu & Donato, 1999). We will then discuss how the method used for estimating the local learning coefficient in (Lau et al., 2024) is related to homogenization.

Homogenization is a process used to understand diffusive processes where the underlying governing structure can have small but rapid variations on small scales. These fluctuations might matter for a diffusing particle on short length/time scales but they should effectively average out at some larger scale. A bit more formally, if we imagine something like a chemical concentration $c^\epsilon(x, t)$ which is diffusing according to the PDE

$$\frac{\partial c^\epsilon}{\partial t} = \nabla \cdot (\mathcal{D}(\frac{x}{\epsilon})\nabla c^\epsilon) \tag{67}$$

where the diffusion $\mathcal{D}$ coefficient varies rapidly when $\epsilon \ll 1$. However, if $\mathcal{D}$ is bounded, then homogenization theory tells us that there is some other function $c^0$ given by $\epsilon \to 0$ such that there is some effective PDE:

$$\frac{\partial c^0}{\partial t} = \nabla \cdot (\hat{\mathcal{D}}(\frac{x}{\epsilon})\nabla c^0) \tag{68}$$

where $\hat{\mathcal{D}}$ is an effective diffusion coefficient which only varies over a much larger scale. This is effectively taking the PDE and averaging out the fluctuations over a particular scale to get something that is easier to model. When performing a homogenization one normally picks a scale that they are "averaging over". This scale can be picked somewhat arbitrarily but making the scale too large or too small can negatively impact how accurately one captures the dynamics of the system. If one takes the scale too small, homogenization is not effective. If one takes the scale too large, you start to ignore how the distribution of fluctuations can change in different areas of the media, leading to an inaccurate theory.

There is a sense in which the local learning coefficient estimation introduced in (Lau et al., 2024) is related to homogenization. For a particular value $w^*$ in the parameter space (which is assumed to be a local minima) and a ball $B_\delta(w^*)$ of radius $\delta$ about $w^*$, they define the learning coefficient estimator as

$$\hat{\lambda}(w^*, \delta) = m\beta[\mathbb{E}_{B_\delta(w^*)}[L_m(w) - L_m(w^*)]] \tag{69}$$

where $w \in B_\delta(w^*)$ and $\beta = \frac{1}{\log m}$. The choice of $\delta$ is effectively the scale over which one is homogenizing, and the estimate of the LLC is akin to the average fluctuation over that area. This is also why when trying to accurately estimate the LLC it is recommended to not make $\delta$ too large.

# F  ROLE OF THE FRACTIONAL DERIVATIVE

## F.1  THE FRACTIONAL DERIVATIVE

The *Caputo fractional derivative*

$$\mathcal{D}_t^\alpha f(t) = \frac{1}{\Gamma(1-\alpha)} \int_0^t \frac{f'(t)}{(t-\tau)^\alpha} d\tau \tag{70}$$

is essentially like a derivative with memory of past derivatives, weighted by a power law decay in time controlled by $\alpha$. To see this, one can consider two extreme cases. First, taking $\alpha \to 0$ you get the total net change of $f(t) - f(0)$. Taking $\alpha \to 1$ you recover something more akin to the "slope" between the time $t$ and the start time. $\alpha$ effectively controls how quickly you ignore the past.

If we want to see how it induces power law subdiffusion consider the linear function $f(t) = at + b$. Assuming $0 < \alpha < 1$, we find

$$\mathcal{D}_t^\alpha f(t) = \frac{a}{\Gamma(2-\alpha)} t^{1-\alpha} \tag{71}$$

So notice that as $\alpha \to 0$ the process becomes more linear, so $\alpha$ controls how "sublinear" the process is.

## F.2  FRACTAL DIMENSION AND FRACTIONAL DERIVATIVE

The relationship between the fractal dimension and the fractional derivative operator has been a subject of investigation for nearly 3 decades, starting with (TATOM, 1995). The authors used numerical simulations to study the relationship between the fractional derivative and the fractal dimensions of particular curves, finding a linear relationship between the order of the fractional operator and the fractal derivative. Since then, extensive theoretical results have been proven for different types of special functions (see (Liang & Su, 2024) for an overview). It was proven in (Songping, 2004) that there is a linear relationship between the Minkowski–Bouligand dimension of the Weierstrass function and the Minkowski–Bouligand dimension of its corresponding fractional calculus. We hypothesize that the fractional derivative in the FFPE for SGD accounts for the change in $\lambda(w)$ as one moves through the parameter space. More concretely:

**Hypothesis F.1** (Shared Slopes). *Let $\alpha(t)$ be the fractional derivative exponent at time $t$. The value of $\alpha(t) \propto \frac{d\lambda(w_t)}{dt}$.*

Since $\alpha(t)$ is effectively a local property of a point that is related to the derivative about the point, this should be unsurprising as the learning coefficient directly describes degenerate directions of the space.

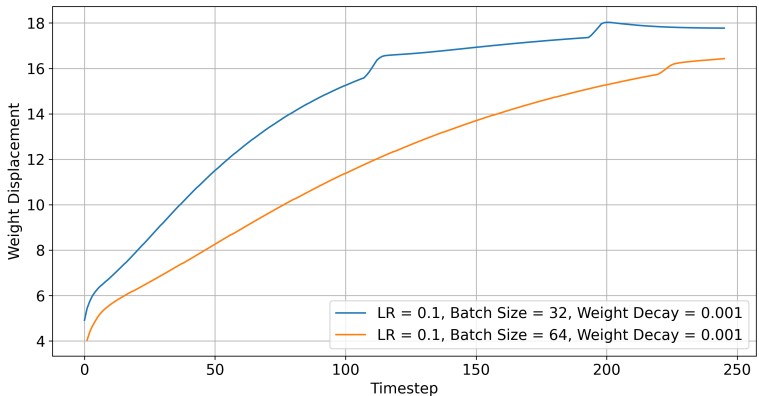

(a) Jump in Weights with SGD Optimizer

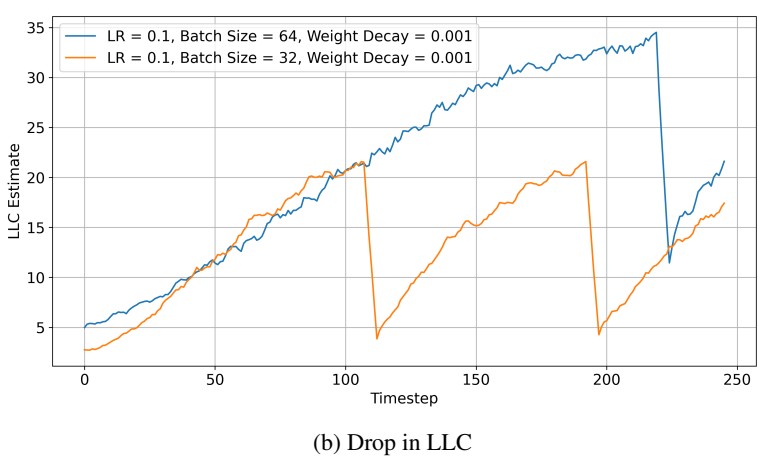

(b) Drop in LLC

Figure 5: The corresponding changes in weight vs. the LLC.

As mentioned in section 3 we see more complex dynamics when we operate in the grokking regime. Experimentally we see (figure 5b) that the appropriate choice of hyperparameters result in sudden large jumps in weight space (and the LLC) when the batch size is sufficiently small. The general sub-diffusive behaviour of these systems is captured by the fractional derivative in time $\mathcal{D}_t^\alpha$. However, the large jumps indicate the need for a fractional derivative in space to fully account for grokking behavior. This could be done by introducing a fractional Laplacian operator to equation 3.1.1 however we don't explore this analytically here. We do note however that the introduction of the space fractional derivative is effectively the same as a Levy noise Langevin equation.

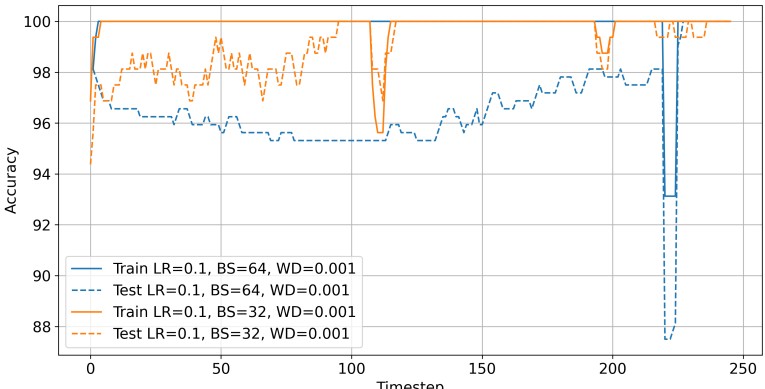

Figure 6: Train/test accuracy over time. Notice that sudden jumps in the accuracy correspond to sudden jumps in the weights and the LLC.

We believe that this is evidence that the concept of stage boundaries and developmental stages introduced in (Wang et al., 2024a) is seemingly a very natural way to discuss stages of learning. They suggest delineating phases of learning by critical points in the (noise mitigated) LLC evolution curve. Our experiments indicate that the rate of change of the local learning coefficient should roughly capture the impact of the time and space fractional derivatives. Discontinuities (or very sharp changes) seemingly account for the action of the spatial fractional derivative, while more stable changes seemingly relate to actions of the time fractional derivative.

## G EXPERIMENT DETAILS

For all experiments, we use the following configuration for the learning coefficient estimation:

Table 3: Hyperparameters for LLC Estimation

| Hyperparameter | Value |
|---|---|
| optimizer_lr | 1e-5 |
| optimizer_localization | 100.0 |
| sampling_method | SGLD |
| num_chains | 1 |
| num_draws | 400 |
| num_burnin_steps | 0 |
| num_steps_bw_draws | 1 |

### G.1 MNIST

To investigate our theory using the MNIST dataset, we take a subset of 10000 images, and create a 50/50 train-test split. We then conduct two different sets of experiments. The first set of experiments are ran on 50 identical models with different random initializations, each trained for 100 epochs with batch size 256 and a learning rate of 0.001. For the other set of experiments we run against a set of 18 different architectures which vary in depth and layer widths, training these for 250 epochs but with the other parameters fixed (a table of architectures is provided in appendix J). We found for our purposes that it is sufficient to use a basic set of hyperparameters for the estimator (appendix J). We compute the LLC every 100 steps, as well as log the displacement of the network from its initial position. We then take the final LLC to be the average over the last 10 estimates. We also performed extensive ablation experiments over optimizers and parameters on MNIST, whose details can be found in appendix H.

## G.2 TINY IMAGENET

Tiny ImageNet (Le & Yang, 2015) is a subset of 200 classes of the full ImageNet dataset, which have been down-sampled to $64 \times 64$ pixels. In our experiments we rescale the images back to $224 \times 224$ as well as apply the standard Imagenet normalization. We conduct experiments on the following models pretrained on ImageNet:

- *ResNet18* (He et al., 2015)
- *ResNet34* (He et al., 2015)
- *VGG11* (Simonyan & Zisserman, 2015)

To conduct our experiments we finetune the above mentioned models after replacing the original output layer with an output layer with 200 neurons. In order to train this network we follow general fine-tuning practices. That is, we freeze the original weights, and fit the new classification head using the Adam optimizer for a maximum 20 epochs with a learning rate of 0.001 and a weight decay of 0.0001, with a batch size of 128. If the loss does not decrease more than 0.0001 over 3 epochs, we stop training, switching to SGD with 0 weight decay and a learning rate of 0.00001 for 2000 steps with batch size 128. We note here that our vision experiments are conducted slightly differently than the language model or MNIST experiments. This was done to test the theory on the "fine-tuning" stage of model development.

## G.3 TINYSTORIES

The TinyStories dataset (Eldan & Li, 2023) was selected as it allows us to explicitly test our theory on late stage training without training models from scratch, but where there are multiple reasonably sized pretrained models which we can compute the LLC for multiple times throughout training. In particular we run experiments on the following models from HuggingFace trained on TinyStories:

- *roneneldan/TinyStories-1M* (hug, b)
- *nickypro/tinyllama-15M* (hug, a)
- *roneneldan/TinyStories-33M* (hug, c)

Each of these models are trained for 1000 steps with a batch size of 16, with a learning rate of 0.00001, with the LLC computed every 100 steps.

## G.4 POSTERIOR CONCENTRATION

To test the posterior concentration predictions we use a simple toy model and dataset where one can reasonably approximate the Bayesian posterior. We use the moons dataset (Pedregosa et al., 2011) with 512 samples, a noise ratio of 0.2, and a batch size of 128. Using this we train a 2 hidden layer ReLU network where each hidden layer has 64 neurons. Each model is trained using SGD with a learning rate of 0.01 for a total of 200 epochs. We do this for 500 random initializations of the network on the same dataset. At the end of training for each model, we compute the LLC, discarding any non-converged training runs. Since our theory is largely about the local posterior, we take the solutions found by SGD and use these to seed SGLD. In particular, since the Bayesian posterior will concentrate around the model with the lowest loss and lowest learning coefficient, we use the SGD samples which have the lowest loss and the lowest LLC. For each run of SGLD we take draw 1000 samples with 200 burn in steps, and 10 steps per sample with a learning rate of 0.00001.

# H MNIST ABLATIONS

Ablations were ran across optimization various hyperparameters for a fully connected network trained on MNIST to better understand the effects hyperparameter choices have on the diffusion characteristics. Experiments are ran for both SGD and Adam to test if the theory is effective for adaptive optimizers. We highlight some of these experiments here.

| Optimizer | $d_s$ (mean) | $d_s$ (std) | $\lambda_{final}$ (mean) | $\lambda_{final}$ (std) | Test Acc (mean) |
|---|---|---|---|---|---|
| adam | 0.4061 | 0.9068 | 3.0957 | 5.7533 | 90.4297 |
| sgd | 7.8165 | 10.2494 | 12.5270 | 11.8393 | 94.0592 |

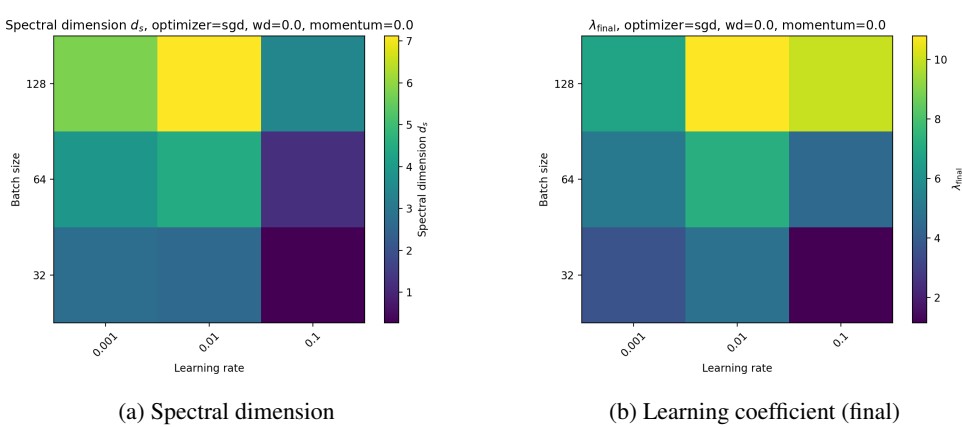

(a) Spectral dimension

(b) Learning coefficient (final)

Figure 7: Learning coefficient and spectral dimension for vanilla SGD with varying batch size and learning rate.

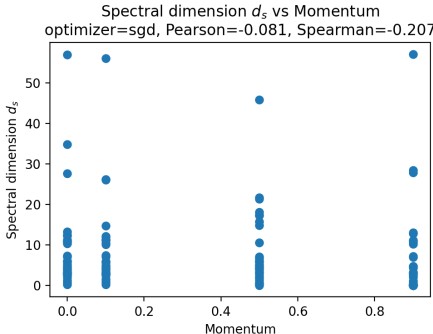

Figure 8: Correlation between momentum and spectral dimension.

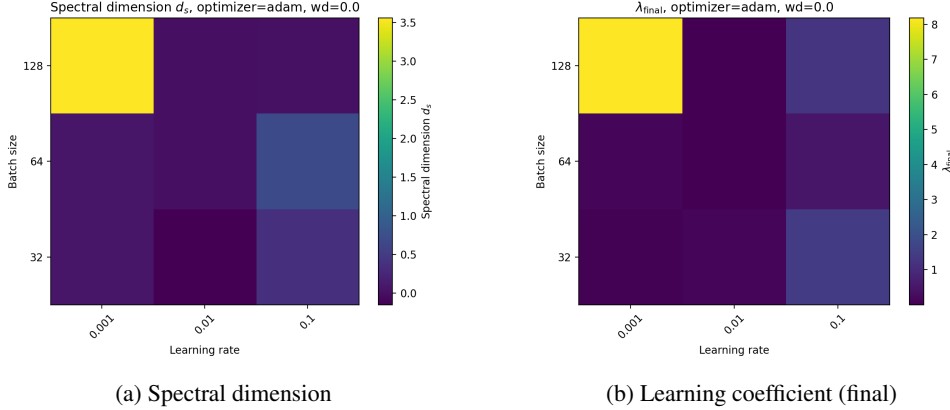

(a) Spectral dimension

(b) Learning coefficient (final)

Figure 9: Learning coefficient and spectral dimension for Adam with varying batch size and learning rate.

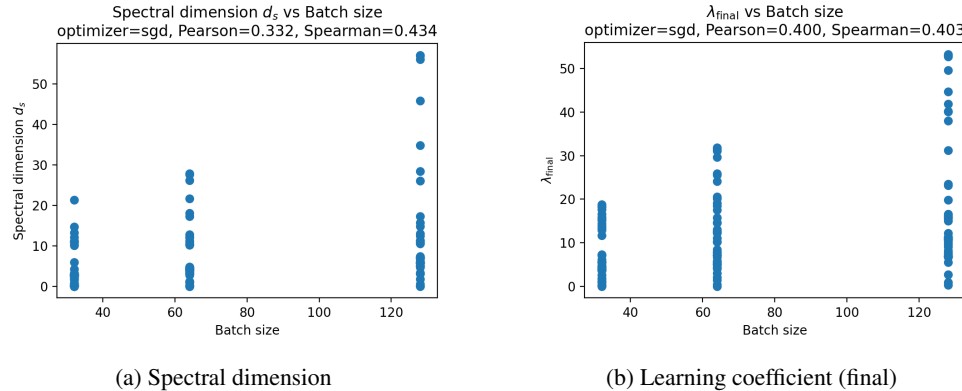

(a) Spectral dimension

(b) Learning coefficient (final)

Figure 10: Correlation of Learning coefficient and spectral dimension for with batch size for SGD

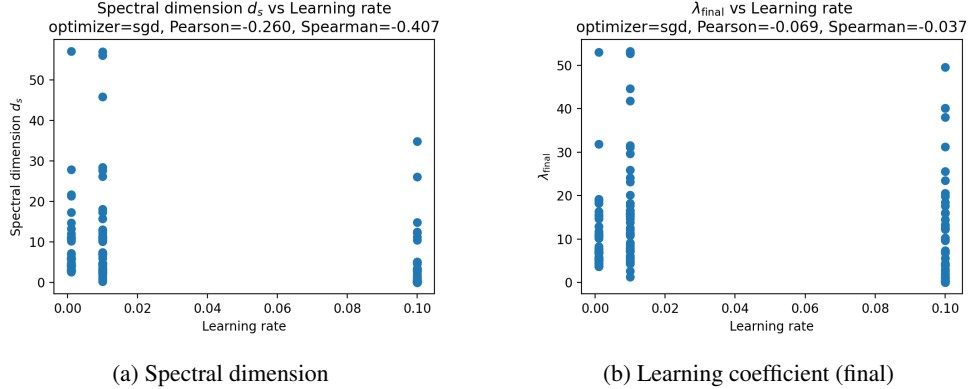

(a) Spectral dimension

(b) Learning coefficient (final)

Figure 11: Correlation of Learning coefficient and spectral dimension with learning rate for SGD

We note that interestingly when using Adam, the spectral dimension seems to have a stronger correlation with performance than the learning coefficient as can be seen in figures 12 and 13. Another interesting phenomena that supports our theory is that there is relatively little correlation between $\lambda$ and the learning rate, but there is relatively substantial correlation between the spectral dimension $d_s$ and the learning rate (figure 11) which aligns well with the theory. We note that the correlation between $\lambda$ and the batch size reflects the sensitivity of the empirical LLC estimator of (Lau et al., 2024) to the batch size.

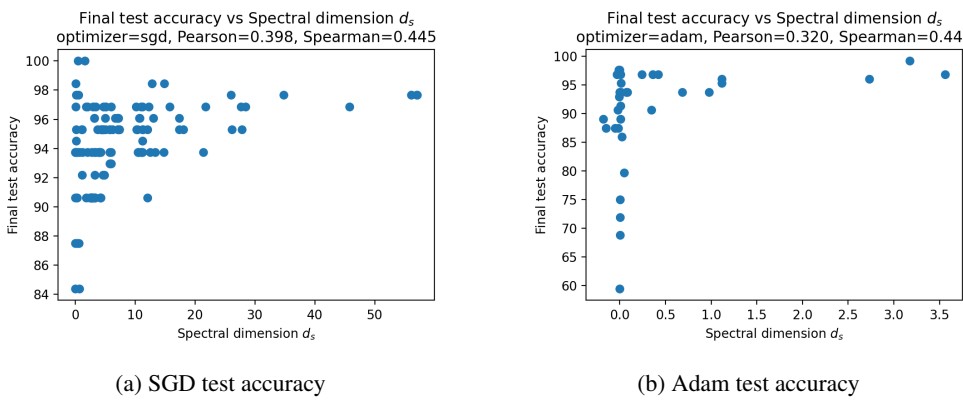

(a) SGD test accuracy

(b) Adam test accuracy

Figure 12: Correlation of spectral dimension with test accuracy for Adam and SGD.

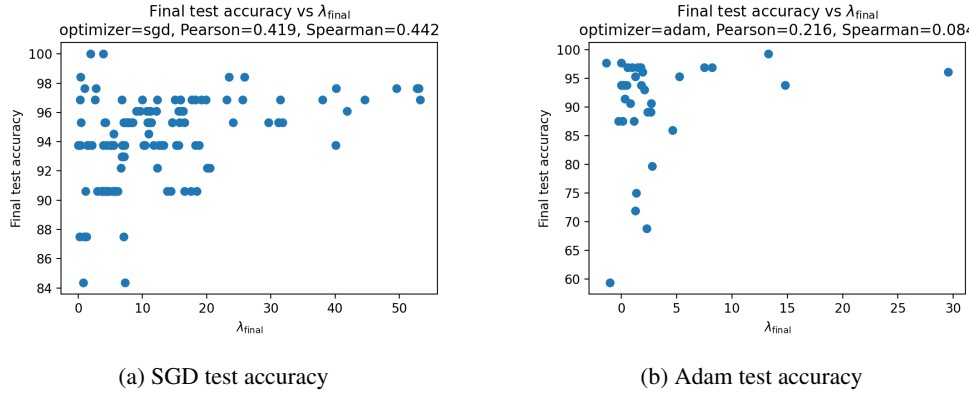

(a) SGD test accuracy

(b) Adam test accuracy

Figure 13: Correlation of the final learning coefficient with test accuracy for Adam and SGD.

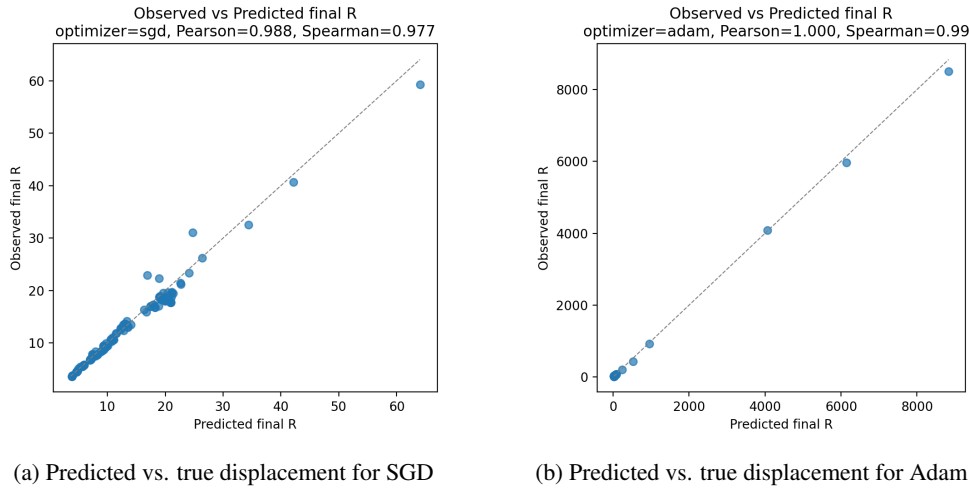

(a) Predicted vs. true displacement for SGD

(b) Predicted vs. true displacement for Adam

Figure 14: Predicted vs. true displacements.

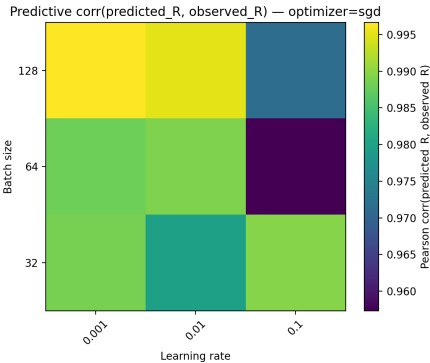

Figure 15: The correlation between the predicted displacement and the true displacement.

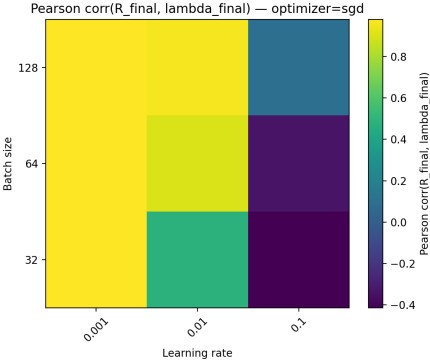

Figure 16: The correlation between the final displacement and the final $\lambda$ vs. batch size and learning rate.

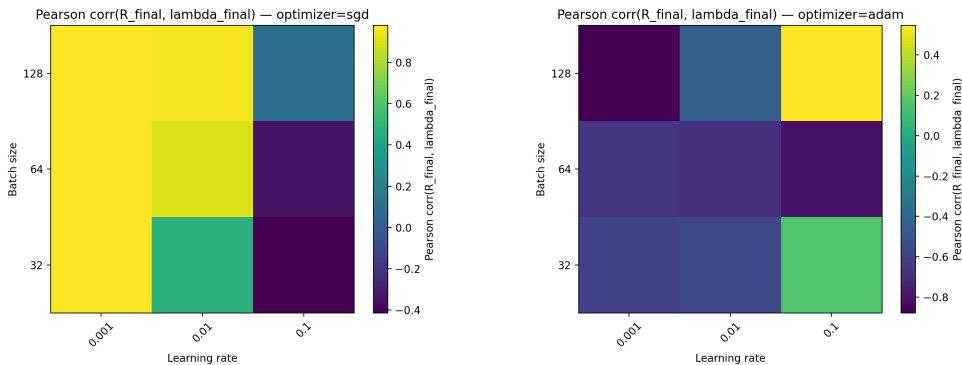

(a) The correlation between the final displacement and the final $\lambda$ vs. batch size and learning rate for SGD.

(b) The correlation between the final displacement and the final $\lambda$ vs. batch size and learning rate for adam.

Figure 17: Correlations between the displacement and learning coefficient.

### H.0.1 SLT AND ADAM

Note that in figure 17 it seems that while the learning coefficient correlates very strongly with SGD, Adam does not. We suggest that this is because the LLC measures a quantity associated with the same Riemannian metric over the data as is used by SGD while adam effectively changes the metric structure via preconditioning. It is well known that variable-metric methods, including adaptive optimizers like Adam, can be interpreted as performing gradient descent in a Riemannian metric defined by a positive-definite preconditioner matrix. In the case of Adam, this preconditioner is the diagonal matrix built from the running average of squared gradients. Recent work has made this connection explicit, showing that Adam can be viewed as an approximate natural-gradient method using a diagonal empirical Fisher information matrix as a data-dependent metric, so the singularity structure of the Adam metric is likely different.

## I ADDITIONAL EXPERIMENTS AND RESULTS

### I.1 DIFFUSION PREDICTION ACCURACY

If the weight diffusion is indeed fractal, we should expect that using the spectral dimension as estimated by equation 17 we should be able to accurately predict the movement of the weights. While this is seen in the runs presented in table 1 they are not presented in the main body for MNIST due to the volume of models trained We can see the histogram of the $R^2$ scores in figure 18. An important thing to note here is that these estimations don't explicitly account for the early super-diffusive

behaviour seen during initial training, so large periods of super diffusion should decrease the accuracy. However, in other settings where we can account for early super-diffusion behaviours, the predictions become nearly exact. An instance of this is in the case of the Tiny ImageNet models where we can explicitly factor out the adaptive training component and simply fit to the SGD component at the end of training (as our theory explicitly cares about late training stages of SGD). This shift in dynamics can be seen in figure 19. However, in instances where we can start with SGD, if the model is already near equilibrium, we can see the sub-diffusive behaviour very early with vanilla SGD. An example of this can be seen in figure 20.

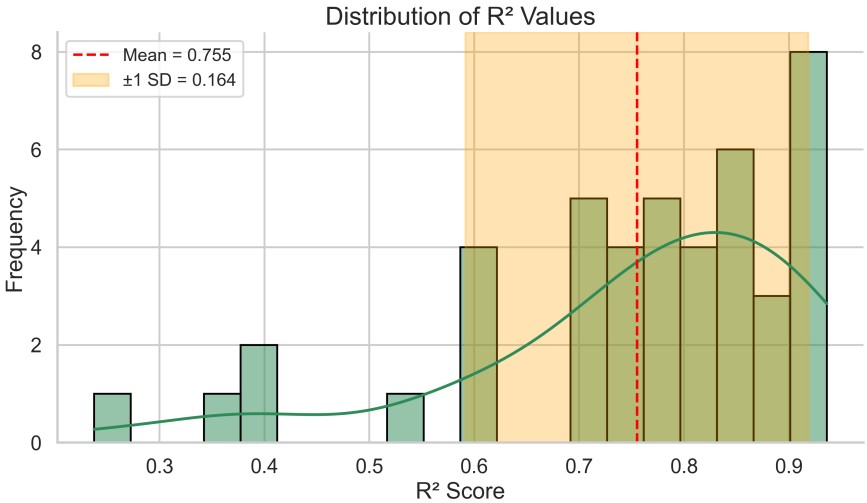

Figure 18: The distribution of the $R^2$ scores for identical MNIST models.

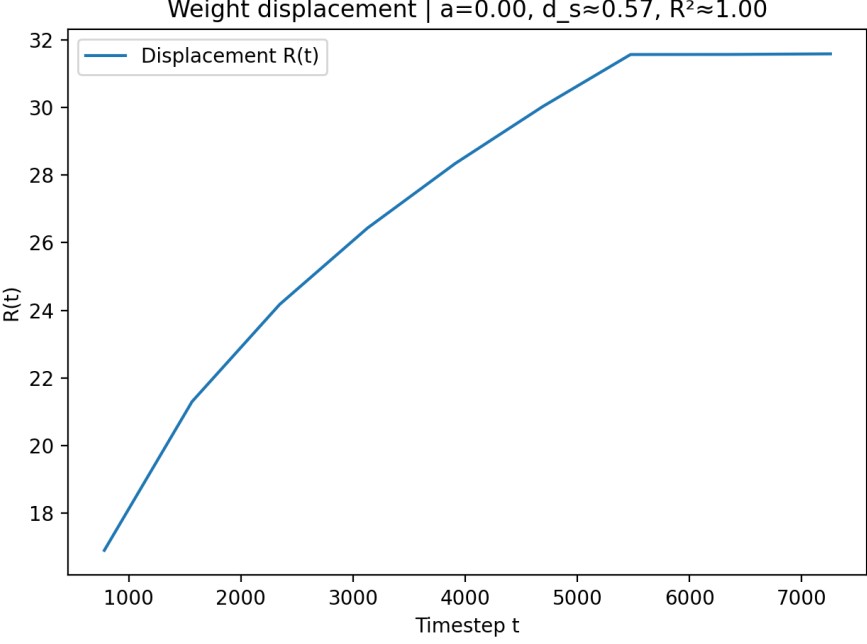

Figure 19: Displacement of the ResNet18 model on Tiny ImageNet. A sharp transition from Adam to SGD can be seen near step 5000, at which point the dynamics become distinctly subdiffusive.

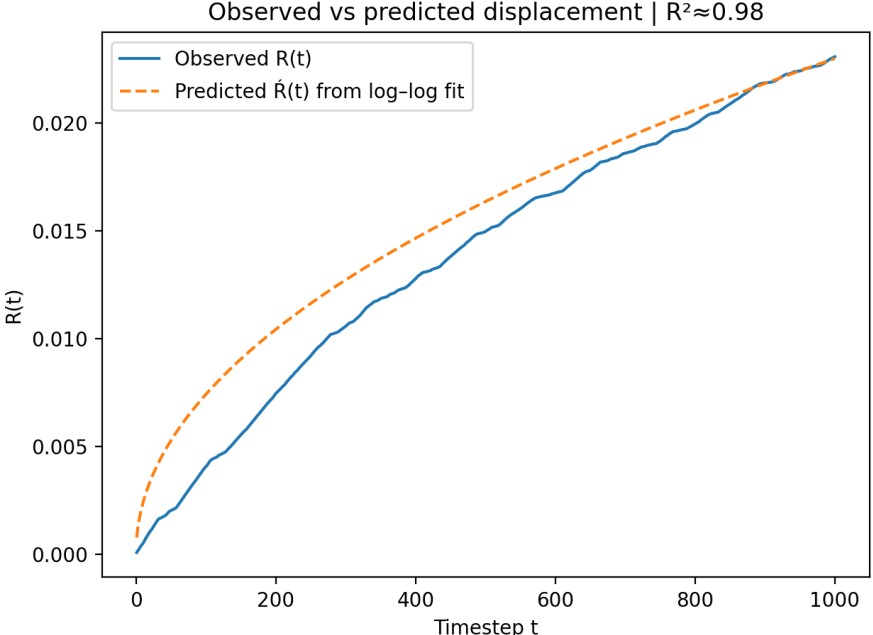

Figure 20: Displacement of the TinyStories-33m model along with the displacement predicted by the theory.

### I.1.1 TINYSTORIES USING ADAM

| Model name | $\lambda$ | $d_s$ | $\alpha$ | $r^2$ |
|---|---|---|---|---|
| TinyStories-1M | 32.05 | 36.6 | 0.57 | 0.99 |
| TinyLlama-15M | 77.02 | 48.06 | 0.31 | 0.97 |
| TinyStories-33M | 40 | 31.95 | 0.39 | 0.98 |

Table 4: Results for language models trained using the Adam optimizer.

Note that while the theory does seem to hold for Adam in some cases, it is less consistent, reflecting the more complex way in which Adam interacts with the geometry.

### I.2 AVERAGE SPECTRAL DIMENSION

We also check the result of corollary 3.3. This can be seen in figure 21

### I.3 CIFAR EXPERIMENTS

We ran additional experiments using various convolutional architectures on CIFAR10 to explore how the super-diffusive component impacts the theoretical results. We find that while these models display stronger super-diffusive behaviours (increasing in intensity with model size) the super-diffusive behaviour attenuates quickly enough that the theoretical results still hold, even when the super-diffusive behaviour is not factored out. Note as well that the runtimes were not as long, so one might expect the super-diffusive component to have a larger impact. In the following we restrict ourselves to a subset of 10000 samples from the CIFAR10 dataset where all models are initialized using Xavier initialization, with 0 bias and ReLU activations.

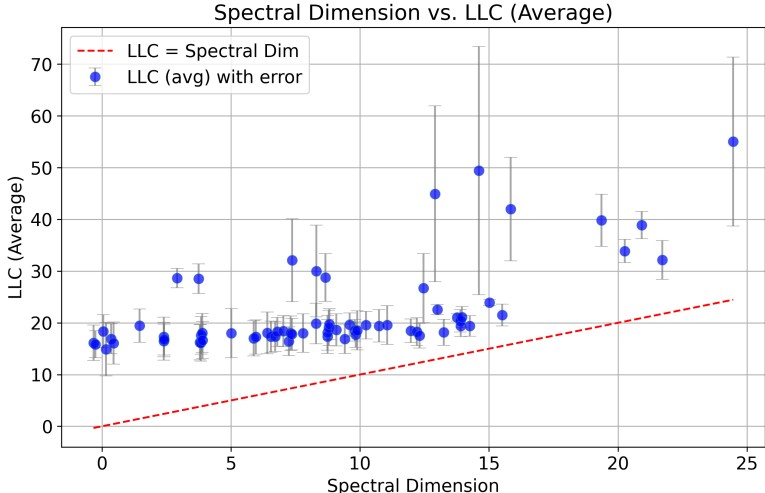

Figure 21: The average LLC vs. the spectral dimension. These results align with corollary 3.3.

Table 5: Spectral Dimension vs. LLC across runs for convolutional architectures

| Num Params | Spect. Dim | LLC | Channels | Batch | Epochs | LR |
|---|---|---|---|---|---|---|
| 1184 | 0.085 | $2.585 \pm 0.779$ | 32 | 64 | 50 | 0.001 |
| 19936 | 3.374 | $5.929 \pm 0.963$ | 32, 64 | 64 | 50 | 0.001 |
| 94304 | 4.887 | $11.829 \pm 1.960$ | 32, 64, 128 | 64 | 50 | 0.001 |
| 372928 | 7.483 | $16.176 \pm 2.175$ | 64, 128, 256 | 64 | 50 | 0.001 |

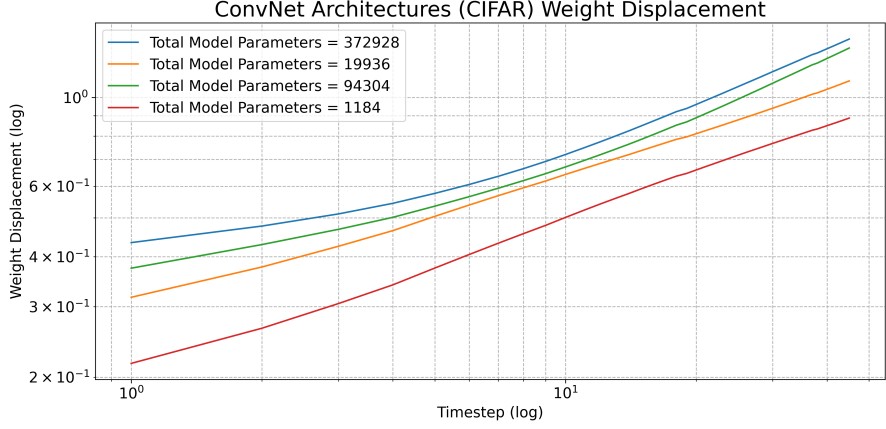

Figure 22: Weight displacement over time across a subset of the CIFAR10 dataset for Convolutional Architectures.

We also investigated the weight displacement for residual architectures. These can be seen in table 6 where the first number is the number of channels in between residual blocks.

Table 6: Spectral Dimension vs. LLC across runs for residual architectures

| Num Params | Spect. Dim | LLC (final) | Channels | Batch | Epochs | LR |
|---|---|---|---|---|---|---|
| 1216 | 0.085 | $2.585 \pm 0.779$ | 32 | 64 | 50 | 0.001 |
| 208448 | 3.481 | $14.729 \pm 4.394$ | 16, 32, 64 | 64 | 50 | 0.001 |
| 167424 | 4.534 | $10.408 \pm 2.127$ | 32, 64 | 64 | 50 | 0.001 |
| 831616 | 5.795 | $24.756 \pm 7.614$ | 32, 64, 128 | 64 | 50 | 0.001 |
| 872640 | 9.652 | $37.731 \pm 8.332$ | 16, 32, 64, 128 | 64 | 50 | 0.001 |

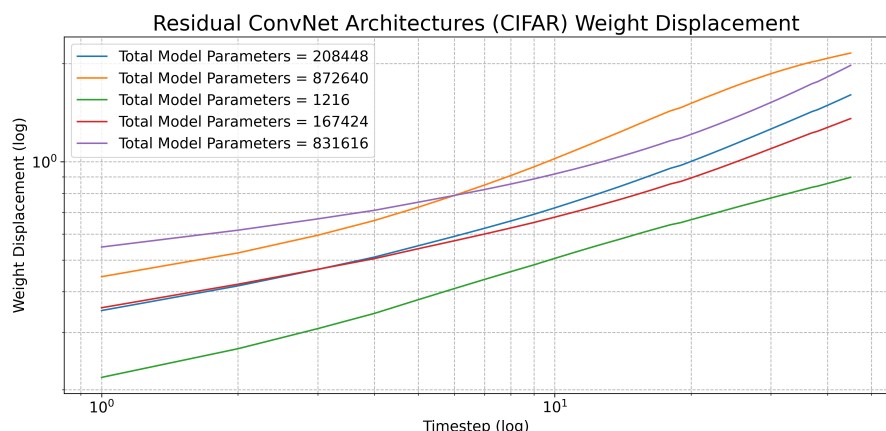

Figure 23: Weight displacement over time across a subset of the CIFAR10 dataset for residual architectures.

If we look at both figures 23 and 22 we can see that larger models exhibit more super-diffusive behavior.

## J   MNIST TABLES

Table 7: Spectral Dimension vs. LLC with hyperparameters for varying model architectures.

| Num Params | Spect. Dim | LLC (Final) | Layers |
| --- | --- | --- | --- |
| 242304 | 2.910 | 30.514 | 784, 256, 128, 64 |
| 125568 | 3.749 | 31.424 | 784, 128, 64, 128, 64 |
| 117376 | 7.373 | 40.128 | 784, 128, 64, 32, 64, 32, 64 |
| 244384 | 8.310 | 38.866 | 784, 256, 128, 64, 32, 16 |
| 127296 | 8.661 | 33.361 | 784, 128, 64, 128, 64, 32 |
| 126112 | 12.458 | 33.385 | 784, 128, 64, 128, 64, 16 |
| 121472 | 12.909 | 61.920 | 784, 128, 64, 32, 64, 32, 64, 32, 64 |
| 118016 | 12.996 | 23.551 | 784, 128, 64, 128 |
| 109184 | 13.763 | 20.324 | 784, 128, 64 1 |
| 125568 | 14.603 | 73.419 | 784, 128, 64, 32, 64, 32, 64, 32, 64, 32, 64 |
| 234752 | 15.021 | 23.299 | 784, 256, 128 |
| 143680 | 15.838 | 51.999 | 784, 128, 64, 128, 64, 128, 64, 32 |
| 141952 | 19.357 | 44.844 | 784, 128, 64, 128, 64, 128, 64 |
| 134400 | 20.268 | 36.134 | 784, 128, 64, 128, 64, 128 |
| 143872 | 20.925 | 41.541 | 784, 128, 64, 128, 64, 256 |
| 244032 | 21.719 | 35.879 | 784, 256, 128, 64, 32 |
| 158336 | 24.465 | 71.359 | 784, 128, 64, 128, 64, 128, 64, 128, 64 |

In table 8 we can see instances where the spectral dimension is negative. This is related to the model having a negative LLC (which occurs generally when there are no near-stable solutions). When negative LLCs occur, it is almost always near initialization where the model also displays super-diffusive displacement (as it is likely moving strongly along a steep gradient). The negative spectral dimension is a remnant of how the spectral dimension is computed in our case. One can reasonably remove occurrences of a negative LLC and perform the estimation which would capture the behavior of the model on the fractal landscape it traverses away from initialization (which is shown to be theoretically valid in appendix B.3). For transparency we do not do this as it illustrates that SGD has complex multifaceted dynamics.

Table 8: Spectral Dimension vs. LLC across runs with different initializations.

| Num Params | Spect. Dim | LLC | Channels/Layers | Batch | Epochs | LR |
|---|---|---|---|---|---|---|
| 110912 | -0.327 | 19.553 | 784, 128, 64, 32 | 256 | 100 | 0.001 |
| 110912 | -0.252 | 18.440 | 784, 128, 64, 32 | 256 | 100 | 0.001 |
| 110912 | 0.048 | 21.552 | 784, 128, 64, 32 | 256 | 100 | 0.001 |
| 110912 | 0.163 | 20.071 | 784, 128, 64, 32 | 256 | 100 | 0.001 |
| 110912 | 0.356 | 19.785 | 784, 128, 64, 32 | 256 | 100 | 0.001 |
| 110912 | 0.455 | 20.136 | 784, 128, 64, 32 | 256 | 100 | 0.001 |
| 110912 | 1.462 | 22.654 | 784, 128, 64, 32 | 256 | 100 | 0.001 |
| 110912 | 2.396 | 21.117 | 784, 128, 64, 32 | 256 | 100 | 0.001 |
| 110912 | 2.401 | 20.125 | 784, 128, 64, 32 | 256 | 100 | 0.001 |
| 110912 | 2.407 | 19.733 | 784, 128, 64, 32 | 256 | 100 | 0.001 |
| 110912 | 3.790 | 19.628 | 784, 128, 64, 32 | 256 | 100 | 0.001 |
| 110912 | 3.824 | 19.682 | 784, 128, 64, 32 | 256 | 100 | 0.001 |
| 110912 | 3.839 | 21.184 | 784, 128, 64, 32 | 256 | 100 | 0.001 |
| 110912 | 3.882 | 21.821 | 784, 128, 64, 32 | 256 | 100 | 0.001 |
| 110912 | 3.899 | 20.301 | 784, 128, 64, 32 | 256 | 100 | 0.001 |
| 110912 | 5.013 | 22.756 | 784, 128, 64, 32 | 256 | 100 | 0.001 |
| 110912 | 5.881 | 20.381 | 784, 128, 64, 32 | 256 | 100 | 0.001 |
| 110912 | 5.968 | 20.591 | 784, 128, 64, 32 | 256 | 100 | 0.001 |
| 110912 | 6.410 | 22.074 | 784, 128, 64, 32 | 256 | 100 | 0.001 |
| 110912 | 6.556 | 20.343 | 784, 128, 64, 32 | 256 | 100 | 0.001 |
| 110912 | 6.713 | 20.250 | 784, 128, 64, 32 | 256 | 100 | 0.001 |
| 110912 | 6.801 | 21.056 | 784, 128, 64, 32 | 256 | 100 | 0.001 |
| 110912 | 7.036 | 21.386 | 784, 128, 64, 32 | 256 | 100 | 0.001 |
| 110912 | 7.240 | 19.237 | 784, 128, 64, 32 | 256 | 100 | 0.001 |
| 110912 | 7.311 | 21.323 | 784, 128, 64, 32 | 256 | 100 | 0.001 |
| 110912 | 7.360 | 20.886 | 784, 128, 64, 32 | 256 | 100 | 0.001 |
| 110912 | 7.794 | 21.724 | 784, 128, 64, 32 | 256 | 100 | 0.001 |
| 110912 | 8.293 | 23.803 | 784, 128, 64, 32 | 256 | 100 | 0.001 |
| 110912 | 8.739 | 20.638 | 784, 128, 64, 32 | 256 | 100 | 0.001 |
| 110912 | 8.749 | 21.454 | 784, 128, 64, 32 | 256 | 100 | 0.001 |
| 110912 | 8.796 | 22.440 | 784, 128, 64, 32 | 256 | 100 | 0.001 |
| 110912 | 8.812 | 22.730 | 784, 128, 64, 32 | 256 | 100 | 0.001 |
| 110912 | 9.081 | 21.760 | 784, 128, 64, 32 | 256 | 100 | 0.001 |
| 110912 | 9.408 | 19.758 | 784, 128, 64, 32 | 256 | 100 | 0.001 |
| 110912 | 9.605 | 21.916 | 784, 128, 64, 32 | 256 | 100 | 0.001 |
| 110912 | 9.795 | 20.633 | 784, 128, 64, 32 | 256 | 100 | 0.001 |
| 110912 | 9.838 | 20.142 | 784, 128, 64, 32 | 256 | 100 | 0.001 |
| 110912 | 9.898 | 22.295 | 784, 128, 64, 32 | 256 | 100 | 0.001 |
| 110912 | 10.230 | 22.213 | 784, 128, 64, 32 | 256 | 100 | 0.001 |
| 110912 | 10.742 | 22.556 | 784, 128, 64, 32 | 256 | 100 | 0.001 |
| 110912 | 11.060 | 23.296 | 784, 128, 64, 32 | 256 | 100 | 0.001 |
| 110912 | 11.966 | 20.767 | 784, 128, 64, 32 | 256 | 100 | 0.001 |
| 110912 | 12.204 | 20.973 | 784, 128, 64, 32 | 256 | 100 | 0.001 |
| 110912 | 12.308 | 19.877 | 784, 128, 64, 32 | 256 | 100 | 0.001 |
| 110912 | 13.253 | 20.701 | 784, 128, 64, 32 | 256 | 100 | 0.001 |
| 110912 | 13.889 | 22.577 | 784, 128, 64, 32 | 256 | 100 | 0.001 |
| 110912 | 13.907 | 21.487 | 784, 128, 64, 32 | 256 | 100 | 0.001 |
| 110912 | 13.952 | 23.129 | 784, 128, 64, 32 | 256 | 100 | 0.001 |
| 110912 | 14.258 | 21.386 | 784, 128, 64, 32 | 256 | 100 | 0.001 |
| 110912 | 15.516 | 23.590 | 784, 128, 64, 32 | 256 | 100 | 0.001 |

## K    COMPUTE RESOURCES

All experiments were run on a single Nvidia RTX A4000 GPU, a single Intel Xeon W-2223 CPU, and 32GB of physical RAM. The individual runs vary greatly in compute times, ranging from $\approx 5$ minutes to $\approx 1$ hour, with the total compute time at $\approx 30$ hours.

## L    LLM USAGE DISCLOSURE

LLMs were used in this work primarily to locate results (formulas, etc) in papers and textbooks.

