# OpenReview forum: "Almost Bayesian: Dynamics of SGD Through Singular Learning Theory"
_ICLR.cc/2026/Conference — ICLR 2026 Poster_

### Official Review · Reviewer_C7eA · 2025-10-24

**Soundness:** 3
**Presentation:** 2
**Contribution:** 3
**Rating:** 6
**Confidence:** 2

**Summary:**

The paper studies the dynamics of SGD algorithms using as tools the singular learning theory. The dynamics of SGD can be approximately modeled using a Langevin SDE, with the exception that the dynamics are super-diffusive during early training and sub-diffusive afterwards, which the authors model using the time fractional Fokker-Planck equation. To capture the local geometry, the local learning coefficient (LLC) and the spectral dimension are utilized, with the LLC essentially acting as a mass dimension and the spectrial dimension determining the reachable area. The resulting stationary state of the Fokker-Planck equation was derived, demonstrating the connection to the Bayes posterior.

**Strengths:**

1. The theoretical analysis is rigorous.
2. Empirical evidence verifies some theoretical claims.

**Weaknesses:**

1. One of the main claim of the paper is the connection of the stationary distribution of SGD dynamics to the Bayes posterior, yet this was not empirically validated.
2. I found limited background provided on singular learning theory; I think this would be beneficial for readers not that familar with singular learning theory.

**Questions:**

I am not familar with the topic of this paper, as such I do not have strong opinions on it. Please see the Weaknesses listed above.

Minor:

Line 065: to related -> to relate

---

> ### Author Response · Authors · 2025-11-20
> **Revisions and Questions**
>
> The authors thank the reviewers for their comments and specifically note that substantially more empirical evaluation has been added, along with other general improvements. Namely:
>
> - Diffusion theory experiments have been ran and reported on both language models (ranging in size form 1M to 33M parameters) as well as experiments on Imagenet vision models.
> - Experimental validation of the Bayesian posterior results, including experimental results explicitly testing the tempered posterior transformation, as well as results showing the effect of different choices of $\xi$.
> - An ablations appendix demonstrating the effects of different hyperparameter choices, for both SGD and Adam.
> - Experiments on language models and MNIST using Adam.
> - Expanded section in the appendix giving background on singular learning theory and LLC estimation.
> - Section in the appendix explaining the spectral dimension in greater depth.
> - An appendix on potential practical applications of the results.
> - General notational cleanup and citation fixes.
>
> In particular, we include extensive experiments on the posterior distribution which we believe addresses the first weakness. Similarly, substantially more introductory material has been added to the appendix.

---

### Official Review · Reviewer_zNv7 · 2025-10-30

**Soundness:** 4
**Presentation:** 3
**Contribution:** 3
**Rating:** 6
**Confidence:** 2

**Summary:**

The paper analyses the behaviour of SGD for NN optimization using concepts from singular learning theory. The key result is a connection between fractal dimension and local behavior of SGD as quantified by a local learning coefficient (LLC), and another one is a link between the diffusive process induced by SGD and a tempered version of the Bayesian posterior, with higher concentration for parts of the parameter space that are easier to reach from an initial condition. The theory is validated using relatively simple empirical experiments that validate the theories related to fractal dimension and LLC.

**Strengths:**

The research question is important: SGD is very widely used and the previous theoretical analysis of its properties relies on simplifying assumptions, some of which are relaxed here, and hence there is clear need for improved understanding. Even though the paper uses somewhat heavy theoretical machinery that may not be accessible for all readers, the key results are understandable for the broader community. I am not an expert with this range of theoretical tools myself, and hence cannot directly evaluate the significance of the contribution, but I find the concrete results convincing.

The empirical experimentation is well justified, with clear explanation of why the experiment is conducted like it was. It is on a small scale, but sufficient for shedding light on the theory.

**Weaknesses:**

It is somewhat difficult to grasp the importance of the spectral dimension, as it is not directly linked to quantities people usually consider. In effect, one needs to be already familiar with Lau et al. (2024); this is not directly a limitation as it is valuable to build on the LLC measure, but the paper would work better as a stand-alone piece if LLC was explained and illustrated a bit more detail here.

The empirical validation is limited to confirming the bounds of Lemma 3.4 and Corollary 3.2 that are both about the spectral dimension. As someone who found Corollary 3.1 a more tangible contact point I would have liked to see something brought from Appendix C to the main paper. I understand the choice though, as that result is more difficult to communicate.

**Questions:**

What can a practitioner learn about your work? Would it be useful to track the spectral dimension in some scenarios and use the information to guide some design choices? What would we gain over tracking LLC? I am very much willing to adapt my score once understanding better the practical significance of the results, but would like to remark that the paper could well be published even as a purely theoretical paper.

The estimated LLC is effectively constant for spectral dimensions between 0 and 15 in both Figure 2 and 3. Why is this? Is the lower bound overly loose for low spectral dimensions, or is this near-constant behavior explained by something else?

---

> ### Author Response · Authors · 2025-11-20
> **Revisions and Questions**
>
> The authors would like to thank the reviewer for their comments and would like to note that the feedback on presentation has been helpful. We highlight some general improvements to the paper here, primarily in the experiments section. Namely:
>
> - Diffusion theory experiments have been ran and reported on both language models (ranging in size form 1M to 33M parameters) as well as experiments on Imagenet vision models.
> - Experimental validation of the Bayesian posterior results, including experimental results explicitly testing the tempered posterior transformation, as well as results showing the effect of different choices of $\xi$.
> - An ablations appendix demonstrating the effects of different hyperparameter choices, for both SGD and Adam.
> - Experiments on language models and MNIST using Adam.
> - Expanded section in the appendix giving background on singular learning theory and LLC estimation.
> - Section in the appendix explainging the spectral dimension in greater depth.
> - An appendix on potential practical applications of the results.
> - General notational cleanup and citation fixes.
>
> ## Weaknesses
> ### LLC vs. Spectral Dimension
> To deal with some of the confusion around the spectral dimension, an appendix has been added to explain what it really is that it does. In effect, the spectral dimension is like a dynamical counterpart to the LLC. An important thing to note is that the theory here is largely to explain the way the LLC impacts training dynamics, and that the behaviour of SGD is largely determined by the singualr structure of the loss landscape. This is seemingly true (in the newly added ablations section, we explicitly check the statistical correlation between final weight displacement and LLC) for SGD but the Adam optimizer displays much stranger behaviour.
>
> **That is, while running ablation studies to address a different question we found that for adaptive optimizers the spectral dimension better predicts model optimizers than the LLC.** This result is shown in a newly added appendix for the MNIST ablations. We do not put it in the main body of the paper as the result has not been explored in much depth but seemingly the way that Adam interacts with the loss landscape is significantly different and the spectral dimension is better able to pick up on this interaction. This represents a potentially important avenue for future work in training dynamics.
>
> ### Weakness: Posterior Validation
> The authors strongly agree with this statement and have thus added a section explicitly validating the posterior claims experimentally on (an admittedly toy) model where we can reasonably estimate the posterior and the set of SGD solutions. We encourage the reviewer to examine these results, and thank them for the reccomendation.
>
> ## Questions
> ### Question 1
> To address questions about practical concerns, we have included an appendix discussing potential applications to optimizer design, learning rate scheduling, calibration, and robustness. For example, the degeneracies of SGD impact its speed of convergence but the same reason for the slow speed of convergence is also the reason SGD tends to generalize better so one might reasonably want to design an optimization scheme that avoids degeneracies when the error is high, but encourages them when the error is low.
>
> While one might not track the spectral dimension itself, it is much cheaper to track the walk dimension (which is a combination of the spectral dimension and LLC) than it is to compute the LLC itself, but for SGD carries much of the same information. The walk dimension can be estimated by simply computing how displacement scales with time. This is likely more practical for very large networks, where computing the LLC throughout training is restrictive. While we would not expect it to fully replace the LLC, the walk dimension is likely more useful as a practical signal to track to trigger other evaluations during training.
>
> ### Question 2
>  Since these are identical networks with the same initialization distribution but different initial values, it is possible that the model is finding similar basins and that these dominate the loss landscape, but the exact path they take to get there is different. This is largely speculation, as the exact mechanism is unknown but it is an interesting observation. The loose bounds property is usually that the estimated LLC tends to be lower, but this would not explain this exact behaviour.

---

> > ### Comment · Reviewer_zNv7 · 2025-11-24
> > **Answer**
> >
> > I confirm that I have checked the updated version and read your response. Overall, the revision addresses my main weaknesses well, and I will keep it in mind during the remaining discussion and final evaluation.
> >
> > The new empirical results in the main paper are indeed helpful, and I appreciate the demonstration on the larger models. While the analysis (a bunch of numbers in Table 1) is necessarily a bit superfifical, it is still useful to indicate that the approach can be used in context of larger models as well, at least by communicating the readers that there are no blockers.
> >
> > Regarding the answer to the 2nd question: I think it would be useful to write 2-3 sentences about this in the paper as well, pointing out the observation and just saying that you do not have a clear explanation for this, possibly with your initial hypothesis as well. Maybe that encourages someone to take a closer look at it.

---

### Official Review · Reviewer_v9AS · 2025-11-01

**Soundness:** 2
**Presentation:** 2
**Contribution:** 3
**Rating:** 6
**Confidence:** 2

**Summary:**

The paper models long-time diffusion dynamics of SGD using the fractional Fokker–Planck equation (FFPE), which can describe sub-diffusive behavior induced by degenerate loss landscapes on a fractal geometry. The authors derived the specific solution of the steady-state distribution of FPPE based on a defined effective diffusion coefficient. They connected it to a tempered Bayesian posterior, modulated by local accessibility constraints determined by the local learning coefficient (LLC). Moreover, they demonstrated that the spectral dimension of SGD is bounded by the fractal dimension, and empirically verified this on MNIST.

**Strengths:**

1. This paper employs fractional SDE and spectral dimension analysis in a coherent framework, which introduces a novel insight to bridge the gap between SGD and Bayesian methods. All lemmas and corollaries are provided with detailed proofs.
2. The idea of modeling anomalous diffusion using the FFPE basis grounded in fractal geometry is reasonable and effective.

**Weaknesses:**

1. The experimental validation only utilizes the small MNIST and CIFAR datasets without larger-scale input, which limits the empirical generality of the proposed inequality $d_{s} \leq \lambda(w(t))$.
2. Experiments use only vanilla SGD without momentum or adaptive variants (Adam, RMSProp). It is unclear whether the proposed fractional diffusion framework can be extended to these more practical optimizers.
3. There is no ablation study about how hyperparameters (e.g., batch size, weight decay, epochs) affect the spectral dimension or LLC evolution.

**Questions:**

1. To assess the empirical fidelity of the “almost Bayesian” claim, can you quantify how closely the steady-state distribution $p_{s}(w)$ matches a true Bayesian posterior of SGLD samples as in Figure 5 (e.g., via KL divergence / Wasserstein distance)?
2. Have you attempted to compute LLC or spectral dimensions on larger or more complex networks (e.g., CNNs on ImageNet)?
3. The fractional Fokker–Planck framework only focuses on sub-diffusion. How would the model extend to super-diffusive or Lévy-like early dynamics, and would this modify the steady-state form?
4. Can you provide any quantitative evidence (e.g., LLC temporal stability plots) supporting the near-stability hypothesis 3.1 during late training?

---

> ### Author Response · Authors · 2025-11-20
> **Revisions and Questions**
>
> The authors thank the reviewer for their thoughtful inputs and comments. In accordance with the provided feedback, we have substantially increased the number of experiments and would encourage the reviewer to reasses previous points. The changes are discussed below as well as answers to the reviewer questions.
>
> ## Weaknesses
> The authors agree with the stated weaknesses and have thus added substantially more experiments to address these. We note these below along with a handful of other changes.
>
> - Diffusion theory experiments have been ran and reported on both language models (ranging in size form 1M to 33M parameters) as well as experiments on Imagenet vision models.
> - Experimental validation of the Bayesian posterior results, including experimental results explicitly testing the tempered posterior transformation, as well as results showing the effect of different choices of $\xi$.
> - An ablations appendix demonstrating the effects of different hyperparameter choices, for both SGD and Adam.
> - Experiments on language models and MNIST using Adam.
> - Expanded section in the appendix giving background on singular learning theory and LLC estimation.
> - Section in the appendix explainging the spectral dimension in greater depth.
> - An appendix on potential practical applications of the results.
> - General notational cleanup and citation fixes.
>
> ## Questions
>
> ### Question 1
> The authors agree and have thus added an experimental section which explicitly probes the relationship computing explicit comparison metrics between the tempered SGD distribution and the Bayesian posterior.
>
> ### Question 2
> Yes, experiments have been added on language models ranging from 1M to 33M parameters as well as various vision models trained on ImageNet.
>
> ### Question 3
> The fractional Fokker-Planck equation can be modified to account for super-diffusion using the time-space fractional Fokker-Planck equation. If the dynamics are super-diffusive for a fixed time window, this does not change the steady state distribution in general. However, if it is super-diffusive through the whole process the steady state distribution can be drastically different or not exist at all. We do not observe this in practice except for in the small batch large learning rate regime on small models.
>
> ### Question 4
> The near stability hypothesis has been removed from the work as it is not particularly important for the main results and seems to detract from the main ideas of the paper, and the room was better used for new experimental results. It was meant to exist as a hypothesis that explained why the estimations of the local learning coefficient did not frequently show negative spikes.

---

### Official Review · Reviewer_psyE · 2025-11-07

**Soundness:** 2
**Presentation:** 2
**Contribution:** 3
**Rating:** 2
**Confidence:** 3

**Summary:**

This paper proposes a theoretical connection between stochastic gradient descent (SGD) dynamics and almost Bayesian behavior, by analyzing the steady-state of SGD through the lens of fractional diffusion and spectral geometry. The authors introduce the concept of local learning capacity (LLC) as an analog of an inverse temperature, and relate it to the spectral dimension of the loss landscape via a family of scaling laws. They provide a mathematical derivation of these relations and test them empirically on toy data and small-scale MNIST experiments.

**Strengths:**

- Ambitious conceptual link between stochastic dynamics, spectral geometry, and Bayesian ideas
- The LLC notion is at least intuitively appealing and might motivate future work on quantifying local "temperature" or effective capacity in SGD
- Some of the derivations are mathematically nontrivial and show that the authors know the literature on stochastic processes and singular learning theory
- The paper is generally readable and nicely typeset, with good figures (especially Fig. 1 for intuition)

**Weaknesses:**

- **Steady-state assumption**: many derivations rely on SGD reaching or approximating a steady-state distribution. The authors mention that this may not actually exist or be reachable in practice, but then they still use it for all the main results. This feels like a big logical gap that should be discussed much more clearly.
- **Notation and definitions**: this part is honestly confusing.
  - V(ε) is first defined as a function of ε (Eq. 5), but later it takes a ball B(w*, ε) as argument (Eq. 7). It’s unclear if that’s the same V or a new one.
  - The dependence on ρ and r is dropped mid-derivation.
  - “walker dimension” vs. “walk dimension”: these seem to be the same thing, just inconsistent terminology.
  - Definition 3.3 really reads like a derived result (it relates quantities already defined before), so calling it a definition doesn’t make sense. It should be a theorem or corollary, or at least justified.
- **Assumptions**: The “near-stability hypothesis” (Hyp. 3.1) is just stated but never tested. Likewise, the use of the Alexander–Orbach relation assumes homogeneity that is unlikely to hold for neural loss landscapes. There’s no discussion of what happens if it fails.
- **Experiments are extremely limited**:
  - Only a toy Moons example (few hundred points) and small MNIST subset are shown.
  - No error bars, no statistical tests, no correlation coefficients... so the claim that LLC correlates with the spectral dimension is just a visual impression.
  - The authors even note that the bound is loose, so it’s unclear if the theoretical quantities actually predict anything.
  - There are no ablations or sensitivity studies (learning rate, batch size, the coarse-graining scale ξ, etc.).
- **Practical relevance unclear**: given the weak and small-scale experiments, it’s hard to see what we should take away practically. The theory could still be interesting, but it would help to discuss possible uses or ways to estimate LLC in large-scale setups.
- **Minor stuff**: lots of typos (e.g., “Kullback-Liebler”, “whcih”) and inconsistent citation style ((double parentheses)) all over the place. Would be good to fix that.

**Questions:**

- Can you clarify the dependence of V on ρ and r in Eqs. 5–7? It currently looks like it depends on those implicitly, but the notation hides it. Also, in Eq. 6, you say λ is defined for an arbitrary choice of a – does that mean λ is independent of a? If yes, can you show why?
- Why is Definition 3.3 a definition and not a theorem? It seems to express a relation between previously defined quantities, so it’s not really definitional. Please clarify or justify.
- The Alexander–Orbach relation is usually derived for homogeneous fractal media. Under what assumptions does it hold here, and how do you know those hold for your loss landscapes?
- How exactly is the scale ξ chosen for D_ξ in Eq. 13? Is it a fixed number, dataset dependent, or tuned? Have you checked sensitivity to it?
- You introduce the “near-stability hypothesis” (Hyp. 3.1). Have you tried to test this empirically (e.g., by measuring distances to nearby minima or metastable states)?
- Could you add at least some quantitative evaluation for the LLC–spectral-dimension relation? e.g., correlation coefficients, confidence intervals, or plots across architectures or datasets. Right now, it’s impossible to tell how strong the relation is.
- Please fix the citation formatting (use \citep and \citet correctly) and typos throughout.

---

> ### Author Response · Authors · 2025-11-20
> **Revisions**
>
> The authors thank the reviewer for their input and comments, and have attempted to remediate highlighted issues accordingly.
>
> ## General
> Before addressing specific concerns, the authors would like to address substantial improvements made to the paper, including substantially more experimental validation. Namely:
>
> - Diffusion theory experiments have been ran and reported on both language models (ranging in size form 1M to 33M parameters) as well as experiments on Imagenet vision models.
> - Experimental validation of the Bayesian posterior results, including experimental results explicitly testing the tempered posterior transformation, as well as results showing the effect of different choices of $\xi$.
> - An ablations appendix demonstrating the effects of different hyperparameter choices, for both SGD and Adam.
> - Experiments on language models and MNIST using Adam.
> - Expanded section in the appendix giving background on singular learning theory and LLC estimation.
> - Section in the appendix explainging the spectral dimension in greater depth.
> - An appendix on potential practical applications of the results.
> - General notational cleanup and citation fixes.
>
> We strongly encourage reviewers who considered the experimental results insufficient to consider the revisions.

---

> ### Author Response · Authors · 2025-11-20
> **Rebuttals**
>
> ## Questions
>
> ### Question 1
> The authors note that the answers to these questions are largely the subject of the seminal textbook on singular learning theory (https://www.cambridge.org/core/books/algebraic-geometry-and-statistical-learning-theory/9C8FD1BDC817E2FC79117C7F41544A3A). More information on SLT has been added to the main text, but many of these results will not be addressed in great mathematical depth here as the answers are straightforward but non-trivial with limit comment. The authors refer in particular to chapters 6 and 7 which cover the singular integral $V(\epsilon)$ and appenidx B of (https://arxiv.org/abs/2308.12108) which explains the impact of $r$. In short, $\rho(w)$ cannot change $\lambda$, and there is some $r$ for the learning coefficient about $w^*$ cannot get any larger. Note that $\lambda$ is indeed independent of $a$. $\lambda$ is given by asymptotic behaviour in $\epsilon$ of the volume $V$. The independence from $a$ can be seen from plugging in asymptotic form of the singular integral $V$.
>
> ### Question 2
> The authors agree and have changed it to a corollary of previous results.
>
> ### Question 3
> While the AO relation was originally derived for homogeneous media, it is a standard result that it can be extended to non-homogeneous media locally, giving a local AO relation. Our use of the asymptotic spectral dimension $d_s^\infty$ introduced in section 3.3.1 is in some sense a modeling ansatz saying that asymptotically there is some spectral dimension which captures the global diffusive behaviour even if the local geometric contraints given by the fractal dimension of the media (the LLC in our case) can change locally.
>
> We have edited the main work to make this distinction clearer.
>
> ### Question 4
> Some systems have a characteristic choice of scale, while for others it is tuned. To address this, we have added specific experiments in the main body showing how changing $\xi$ impacts the predictive accuracy of the theory.
>
> ### Question 5
> The near stability hypothesis has been removed from the work as it is not particularly important for the main results and seems to detract from the main ideas of the paper, and the room was better used for new experimental results. It was meant to exist as a hypothesis that explained why the estimations of the local learning coefficient did not frequently show negative spikes. This fact implies the near stability hypothesis indirectly.
>
> ### Question 6
> The spectral dimension is computed from the LLC and the displacement by definition. That is to say, **given an LLC and the displacement of the weights, the spectral dimension is a fixed quantity given by solving the relevant equation outlined in the experiment section**. The spectral dimension is exactly the quantity that transforms the LLC into the walk dimension. In a sub-diffusive system the spectral dimension must be less than the LLC since otherwise the sytem would not be sub-diffusive. What is instead of statistical relevance here is the correlation between the LLC and the displacement. A figure showing this was added to the main body.
>
> ## Addressing Weaknesses
>
> ### Weakness 1: Steady State Assumption
> While SGD is known under mild assumptions to reach an approximate steady state, the authors agree that this was not made clear enough. As such, it is addressed more explicitly in the main body.
>
> ### Weakness 4: Loose Bounds
>
> We note however that the "loose bounds" is not particularly problematic in our case. The loose bound in the learning coefficient is simply that the local learning coefficient estimator is much more likely to give a slightly lower learning coefficient than a higher one, which is in fact an indicator that our theory is somewhat noise robust.
>
> The other "loose bound" comes from some MNIST and CIFAR experiments where early in training the model can exhibit super-diffusion for some initial time before settling into sub-diffusive behaviour. We don't explicitly factor this out for the MNIST experiments (as it mostly occurs outside the regime we explicitly study). However, we do factor this out in the ImageNet model case, where we find after factoring out the initial super-diffusion, the displacement as predicted by the theory is nearly exact.
>
> ## Conclusion
> The authors would like to thank the reviewer for their time and comments. We would encourage the reviewer to re-examine the work given the added experiments and theoretical clarification.

---

### Meta-Review · Area_Chair_BiML · 2026-01-07

**Summary:**

The paper develops a theoretical framework connecting the long-time dynamics of stochastic gradient descent (SGD) to Bayesian inference by modelling SGD as anomalous diffusion on a degenerate (fractal-like) loss landscape. Using tools from singular learning theory (SLT) and fractional Fokker–Planck equations, the authors argue that late-stage SGD dynamics are sub-diffusive due to loss-surface singularities. Central to the framework is the local learning coefficient (LLC), which quantifies local degeneracy and plays a role analogous to an inverse temperature or local dimension controlling accessibility of parameter space.

The main theoretical result is that under assumptions of approximate steady-state behavior, the stationary distribution of SGD corresponds not to the exact Bayesian posterior, but to a tempered posterior that is modulated by local accessibility constraints induced by the geometry of the loss landscape. The authors further relate the LLC to the spectral dimension and walk dimension of the induced diffusion process, deriving bounds and scaling relations that link geometry, diffusion, and stationary distributions. Empirically, the original submission validated these claims primarily on toy problems and small-scale MNIST/CIFAR experiments.

In response to reviewer feedback, the authors substantially expanded the experimental section in later revisions, adding results on language models (up to ~33M parameters), ImageNet-scale vision models, ablations over hyperparameters, experiments with Adam, and explicit empirical comparisons between SGD stationary distributions and Bayesian posteriors. Additional appendices were added to explain SLT, spectral dimension, LLC estimation, and potential practical implications.

**Reviewer Concerns:**

Initial reviews were mixed, leaning negative. The major concerns are as follows:

- Theory (raised by psyE, v9AS, C7eA): Several reviewers questioned the reliance on a steady-state distribution of SGD, noting that such a state may not exist or may be unreachable. Additional concerns included heavy assumptions (e.g. Alexander-Orbach relation, local homogeneity), unclear hypotheses (near-stability), and confusion over definitions and notation. In response, the authors clarified the steady-state assumption, highlighting that it likely holds for SGD (I agree); removed the near-stability hypothesis; fixed definition issues; and provided additional theoretical background and clarifications in Appendices.

- Numerical Experiments (raised by psyE, v9AS, C7eA): Reviewers criticized the initial experimental validation as too small-scale, mostly qualitative, lacking statistics, ablations, optimizer diversity, or quantitative posterior comparisons. Claims of correlation were viewed as visually suggestive rather than demonstrated. In response, the authors added larger-scale experiments involving ImageNet and language models up to 33M parameters; ablation studies; experiments with Adam; and explicit quantitative comparisons between SGD steady-state distributions and Bayesian posteriors. This is the most successfully addressed category, with revisions directly targeting reviewer concerns and significantly strengthening empirical credibility.

- Bayesian posterior claim (raised by psyE, v9AS, C7eA): A major concern was that the paper’s headline claim was not empirically validated, and it remained unclear how close the SGD distribution actually is to a Bayesian posterior. The authors agreed with this sentiment and added experiments explicitly comparing SGD distributions to SGLD Bayesian posteriors using quantitative metrics. They also demonstrated several effects of posterior tempering. This concern has been addressed.

- Clarity (raised by psyE, zNv7, C7eA): Reviewers highlighted confusing notation, inconsistent terminology, insufficient explanation of SLT concepts, and unclear motivation for quantities like the spectral dimension. The authors performed a notational cleanup, citation fixes, added extensive background appendices, expanded explanations of spectral dimension and LLC, and clarified the role of spectral dimension vs LLC. Reviewer zNv7 acknowledged that the clarity has improved, although the paper still benefits from prior familiarity with LLC.

- Significance (raised by psyE, zNv7): Reviewers questioned what practitioners should do with these insights and whether tracking the spectral dimension or LLC offers value beyond existing diagnostics. The authors added a new appendix on potential applications (optimizer design, scheduling, robustness) and suggested tracking walk dimension as a cheaper proxy. These responses are sufficient for a theory-forward paper.

To the authors, I would request that in the future, any changes made in a revision to OpenReview should be marked with a different color to ensure that the changes have actually been made appropriately.

**Reviewer Scores:**

Reviewers that had the opportunity to respond acknowledged the many improvements to the document; indeed, Reviewer zNv7 directly implied that they would be willing to raise their score at a later time. The most negative reviewer, Reviewer psyE, leans heavily into the criticisms about the steady-state assumption. I do not find this to be as much of an issue, but it is unclear whether the author response would have resulted in a score increase. Regardless, the sheer number of positive changes have certainly improved the paper beyond its initial offering which was already deemed borderline. With further discussion, the scores may have improved to a clear accept. With that in mind, I give a tentative recommendation for acceptance.

---

### Decision · Program_Chairs · 2026-01-26

Accept (Poster)